# Stereotyped behavioral maturation and rhythmic quiescence in *C. elegans* embryos

Evan L Ardiel[1,2]*[†], Andrew Lauziere[3,4][†], Stephen Xu[3], Brandon J Harvey[3,5], Ryan Patrick Christensen[3], Stephen Nurrish[1,2], Joshua M Kaplan[1,2], Hari Shroff[3,5,6]

[1]Department of Molecular Biology, Massachusetts General Hospital, Boston, United States; [2]Department of Neurobiology, Harvard Medical School, Boston, United States; [3]National Institute of Biomedical Imaging and Bioengineering, National Institutes of Health, Bethesda, United States; [4]Department of Mathematics, University of Maryland, College Park, United States; [5]Fellows Program, Marine Biological Laboratory, Woods Hole, United States; [6]Janelia Research Campus, Howard Hughes Medical Institute, Ashburn, United States

**\*For correspondence:**
ardiel@molbio.mgh.harvard.edu

[†]These authors contributed equally to this work

**Competing interest:** The authors declare that no competing interests exist.

**Abstract** Systematic analysis of rich behavioral recordings is being used to uncover how circuits encode complex behaviors. Here, we apply this approach to embryos. What are the first embryonic behaviors and how do they evolve as early neurodevelopment ensues? To address these questions, we present a systematic description of behavioral maturation for *Caenorhabditis elegans* embryos. Posture libraries were built using a genetically encoded motion capture suit imaged with light-sheet microscopy and annotated using custom tracking software. Analysis of cell trajectories, postures, and behavioral motifs revealed a stereotyped developmental progression. Early movement is dominated by flipping between dorsal and ventral coiling, which gradually slows into a period of reduced motility. Late-stage embryos exhibit sinusoidal waves of dorsoventral bends, prolonged bouts of directed motion, and a rhythmic pattern of pausing, which we designate slow wave twitch (SWT). Synaptic transmission is required for late-stage motion but not for early flipping nor the intervening inactive phase. A high-throughput behavioral assay and calcium imaging revealed that SWT is elicited by the rhythmic activity of a quiescence-promoting neuron (RIS). Similar periodic quiescent states are seen prenatally in diverse animals and may play an important role in promoting normal developmental outcomes.

## Editor's evaluation

Embryonic behavior is a widespread phenomenon that remains poorly understood in any model system. Recent developments in imaging, genetic, and computational tools allow for unprecedented analyses of motor behaviors, and the patterns of neuronal activity underlying embryonic development. Here, Ardiel et colleagues establish the roundworm *C. elegans* as a powerful system to study the developmental trajectories corresponding to embryonic behavior and to provide mechanistic insight into how late-stage bouts of activity are modulated.

## Introduction

The set of synaptic connections assembled during development form a physical substrate for brain activity, behavior, and cognition. Diverse model systems have been deployed to identify the molecular components essential for initiating this process, from neuronal differentiation to synaptogenesis. In

parallel, postnatal behaviors have been studied from genes to cells to circuits. By comparison, few datasets probe the earliest emergence of functional neural circuitry. New principles are likely to be discovered through focused analysis of nascent neuronal circuits and the behaviors they elicit. Indeed, the structural remodeling of synapses peaks early and becomes less dynamic with age and the maturation of some circuitry is restricted to specific stages (critical periods) early in development *Hensch, 2004*.

In 1695, Antonie van Leeuwenhoek expressed his amazement at first seeing motility in unborn mussels, writing that the graceful rotations were beyond comprehension *Preyer, 1885*. Careful observation has since uncovered details of embryonic behavior across phylogeny. Studies of the chick embryo were particularly influential. Here, it was established that rhythmic motility was mediated by stereotyped spontaneous activity in neuromuscular circuitry *Hamburger, 1963*. Behavioral classification has traditionally relied on expert observers; however, new tools for data-driven analysis of rich behavioral recordings are accelerating progress in ethology and systems neuroscience *Berman, 2018*; *Brown and de Bivort, 2018*; *Datta et al., 2019*; *Pereira et al., 2020*. Applying these tools to embryos has the potential to illuminate how the brain assumes control of motor output. What are the first behaviors that embryos exhibit? Is there a stereotyped developmental progression for embryonic behaviors? Is this progression dictated by the timing of ongoing neurodevelopment (i.e. circuit wiring)? Do embryonic circuits encode behaviors by mechanisms distinct from those found in mature, post-natal circuits?

To begin addressing these questions, here we build a comprehensive picture of embryonic behavior in the microscopic roundworm *Caenorhabditis elegans. C. elegans* offers several advantages for investigating the emergence of functional neuronal circuitry and behavior. Embryos are small ($\approx 50x30x30\mu m^3$) and transparent, enabling in vivo imaging of cell migration, neurite growth, synaptogenesis, neuronal activity, and behavior during the mere ≈14 hr from fertilization to hatching *Shah et al., 2017*; *Barnes et al., 2020*; *Heiman and Shaham, 2009*; *Fan et al., 2019*; *Christensen et al., 2015*; *Moyle et al., 2021*; *McDonald et al., 2020*; *Sengupta et al., 2021*; *Ardiel et al., 2017*; *Hall and Hedgecock, 1991*; *Baris Atakan et al., 2020*; *Bayer et al., 2022*. In addition, *C. elegans* boasts a sophisticated molecular genetic toolkit, a complete neuronal wiring diagram, and a comprehensive atlas of single cell gene expression across embryogenesis *White et al., 1986*; *Cook et al., 2019*; *Witvliet et al., 2021*; *Packer et al., 2019*. *C. elegans* has proven a powerful system for analyzing circuits and behavior; however, virtually all such studies have focused on post-embryonic stages, including 'cradle to grave' behavioral recordings *Stern et al., 2017*; *Churgin et al., 2017*. In embryos, cell movement has been fully tracked from the first cell division up to the first muscle contraction, which occurs at about 430 minutes post-fertilization (mpf) *Bao et al., 2006*; *Giurumescu et al., 2012*; *Kumar et al., 2016*. Tracking behavior beyond this point is complicated by the embryo's rapid movements and entangled body postures. At hatching, the 222-cell larval nervous system already supports directed locomotion and even learning and memory *Hong et al., 2017*. Thus, a functional motor program emerges over what is arguably the least documented 6 hr of the worm's life (430–800 mpf).

When the nervous system begins generating behavior is currently unknown. Pre-synaptic proteins are first localized in the worm's 'brain', a large circumferential bundle of axons called the nerve ring, less than an hour before the first muscle contraction *McDonald et al., 2020*. However, these pre-synaptic proteins exhibit high diffusional mobility at this stage *McDonald et al., 2020* and it is unclear when nerve ring synapses become functional. Mutants lacking essential muscle genes arrest as twofold embryos, indicating that muscle contractions are required for elongation of the embryo's body *Williams and Waterston, 1994*. Elongation is unaffected in mutants deficient for neuronal function, implying that initial muscle contractions do not require neuronal input. In a developmental series of electron micrographs, pre-synaptic vesicle clusters at body wall neuromuscular junctions are first observed a couple of hours after the onset of movement (≈550 mpf) *Durbin, 1987*. In late-stage embryos, backward locomotion correlates with activity of backward command interneurons *Ardiel et al., 2017*, suggesting the existence of a functional motor program prior to hatching. Here, we documented all postural transitions for the final few hours of embryogenesis (530 mpf to hatching). We find that embryos exhibit a stereotyped pattern of behavioral maturation, including a period of rhythmic quiescence. This rhythmic quiescence is elicited by the RIS neuron, which promotes sleep-like behaviors in larvae and adults *Turek et al., 2016*; *Wu et al., 2018*. Rhythmic sleep-like states are a highly conserved feature of prenatal behavior in other animals *Balaban et al., 2012*; *Pereanu et al.,*

*2007*; *Okai et al., 1992*; *Szeto and Hinman, 1985*; *Corner, 1977*. C. elegans offers a powerful system for investigating this phenomenon.

## Results

### Building embryo posture libraries

Building posture libraries to investigate embryo behavior proved challenging for several reasons. First, embryos are susceptible to photodamage, requiring us to carefully calibrate illumination dose, acquisition speed, spatial resolution, and signal-to-noise ratio. Second, at the low illumination dose required for sustained embryo imaging, fluorescent labels used to visualize posture are dim, necessitating optimized segmentation schemes to detect them. Third, the segmented labels must be tracked accurately, a task complicated by imperfect cell detections and abrupt embryo movements.

To assess posture, we imaged ten pairs of skin cells (termed seam cells) that run along the left and right sides of the *C. elegans* embryo. We defined posture as the joint identification of all seam cell nuclei in an image volume. Seam cells expressing nuclear localized GFP were imaged by light-sheet fluorescence microscopy, which allows rapid optical sectioning with minimal photodamage. With single view imaging (iSPIM) *Wu et al., 2013*; *Kumar et al., 2014*, we recorded volumes at 3 Hz for more than 4.5 hr (*Figure 1a and b*; *Figure 1—video 1*). Imaging did not affect the timing of the Q/V5 cell division (≈695–725 mpf) nor hatching (<820 mpf) and did not induce detectable photobleaching. Consequently, we conclude that this imaging protocol did not interfere with normal embryonic development.

To track posture, seam cell nuclei must first be accurately detected (i.e. distinguished from each other and background). Using image volumes with manually annotated seam cells, we compared the performance of several image segmentation methods. The best performance was obtained using a 3D convolutional neural network *Ronneberger et al., 2015*; *Çiçek et al., 2016*, although the Laplacian of Gaussians *Lindeberg, 1998*; *Lowe, 2004*; *Tinevez et al., 2017*, a traditional method for blob detection, achieved the same level of precision with only a slightly lower recall across the test set (*Table 1*).

After detection, we evaluated methods for enabling comprehensive tracking of nuclear locations across all image volumes. Different multiple object tracking (MOT) methods trade computational complexity for modeling capability https://doi.org/10.1016/j.artint.2020.103448. The first, and simplest, MOT strategy considered was the Global Nearest Neighbor (GNN) method. In GNN, nuclei are treated as independent objects, and are tracked by minimizing frame-to-frame object displacement. However, GNN often failed when nuclear trajectories intersected or when the detection set contained false positives, coordinates erroneously counted as nuclei centers. GNN can be augmented to improve performance. For example, dynamical physical modeling (e.g. Kalman Filtering; *Kalman, 1960*) can be incorporated to predict object trajectories. Multiple hypothesis tracking (MHT) *Reid, 2015*; *Blackman, 2018*; *Cox and Hingorani, 1996a* leverages this strategy and considers multiple future frames to distinguish between competing object assignments in the current frame. MHT is considered a state-of-the-art MOT method; however, MHT is most effective when imaging is rapid enough that object trajectories appear smooth. Unfortunately, for late-stage embryos, a volumetric imaging rate of 3 Hz (selected to preserve embryo health) does not reliably produce smooth trajectories.

Seam cells have fixed anatomical positions that define a flexible lattice. This anatomical constraint causes seam cells to move in a correlated manner. We reasoned that MHT could be improved by incorporating these correlations. To test this idea, we developed Multiple hypothesis hypergraph tracking (MHHT) *Lauziere et al., 2021*. MHHT extends MHT by using empirically derived covariances (*Figure 1c*) to compute the cost of competing cell assignment hypotheses. MHHT also excludes hypotheses that generate physically impossible body contortions (i.e. body segments passing through one another). MHHT then identifies the cost minimizing posture of $K$ sampled hypotheses over $N$ future frames (*Figure 1d*). MHHT should improve tracking compared to strategies where nuclei are assumed to move independently.

To compare the efficacy of MHHT to simpler tracking models, we performed simulations on a test embryo (51,533 image volumes) for which seam cell positions were manually annotated. Using the annotated nuclear locations, MHHT ($K$=5, N=5) incorrectly assigned at least one seam cell (and thus misidentified posture) in 1.43% of image volumes, which was significantly less than the error rate

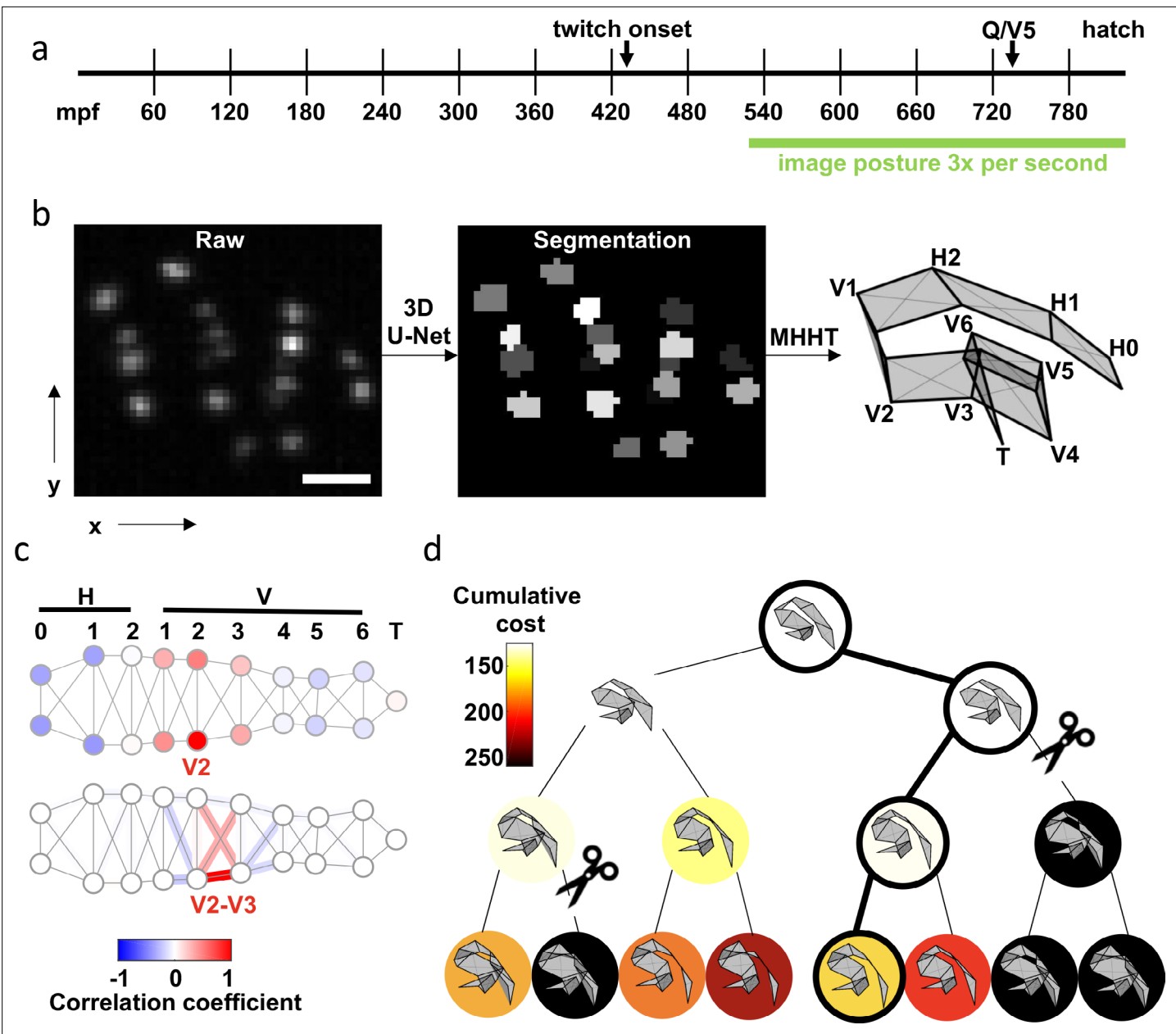

**Figure 1.** Tracking seams cells with Multiple Hypothesis Hypergraph Tracking (MHHT). (**a**) A timeline of embryo development is shown with the imaging window highlighted. mpf: minutes post fertilization. (**b**) The scheme utilized for seam cell identification is shown. Left: A representative image of hypodermal seam cell nuclei (XY maximum intensity projection). Scale bar, 10 $\mu m$. Middle: Output of the 3D U-Net segmentation. Right: Seam cell identifications made by MHHT. See *Figure 1—video 1*. (**c**) To illustrate the interdependence of seam cell movement, correlation in X-axis (labelled in b) motion of the V2 seam cell with all other seam cells is shown (top). To illustrate local constraints on the distance between neighboring seam cells, correlation of V2-V3 distance with indicated edge lengths is shown (bottom). See *Figure 1—figure supplement 1* for full correlation matrices. (**d**) A posture assignment search tree with width $K$=2 and depth $N$=3. Physically impossible body contortions (self-intersections) are pruned (scissors). The path accruing the least cost is highlighted (outlined circles).

The online version of this article includes the following video and figure supplement(s) for figure 1:

**Figure supplement 1.** Correlations used by MHHT.

**Figure 1—video 1.** An embryo at 540 mpf expressing GFP in hypodermal seam cell nuclei.

https://elifesciences.org/articles/76836/figures#fig1video1

**Table 1.** Seam cell nuclei detection results on a held out test set.

The variance in apparent size, shape, proximity, and intensity patterns make seam cell nuclei detection challenging. A held-out test set of image volumes was annotated by an expert; centers of seam cell nuclei were used as reference to compare automatic detection methods. A variety of methods were compared, with recent deep learning based methods yielding the most accurate detections, on average (across 46 held-out test image volumes). The large kernel 3D U-Net trained on both the dice coefficient (Dice) and binary cross-entropy (BCE) yields both the highest precision and recall. However, even the 3D U-Net cannot reliably detect all seam cell nuclei in an image volume.

| Model | Precision | Recall | F1 | Citation | Implementation |
|---|---|---|---|---|---|
| IFT-Watershed | 0.81 | 0.80 | 0.80 | *Falcão et al., 2004* | *Lombardot, 2017* |
| LoG-GSF | 0.95 | 0.90 | 0.92 | *Lindeberg, 1998*; *Lowe, 2004* | *Tinevez et al., 2017* |
| Wavelet | 0.88 | 0.86 | 0.87 | *Olivo-Marin, 2002* | *de Chaumont et al., 2012* |
| Mask-RCNN | 0.93 | 0.89 | 0.91 | *He et al., 2017* | *Waleed, 2017* |
| 3D U-Net Dice | 0.94 | 0.91 | 0.92 | *Çiçek et al., 2016* | *Chollet, 2021* |
| 3D U-Net Dice/BCE | 0.95 | 0.92 | 0.93 | *Çiçek et al., 2016* | *Chollet, 2021* |
| Stardist 3D | 0.91 | 0.88 | 0.89 | *Weigert et al., 2020* | *Weigart, 2021* |

obtained with GNN tracking (2.01% of volumes, $K$=1, N=1, $p < 5.1\ e-13$). Higher error rates were observed when seam cells were detected using the modified 3D U-Net; however, MHHT (5.25% of volumes, $K$=25, N=2) still outperformed GNN (5.88% of volumes, $K$=1, N=1, $p <1.02\ e-5$). MHHT also has a feature designed to anticipate tracking failures. When the lowest assigned cost crosses a user-defined threshold, MHHT prompts the user to manually edit cell assignments before resuming automated tracking. The MHHT code is freely available at https://github.com/lauziere/MHHT; *Lauziere et al., 2021*.

Collectively, these methods (iSPIM, 3D U-Net segmentation, and MHHT) comprise a semi-automated pipeline for documenting embryo postures. Using these methods, we generated a comprehensive catalogue of postural transitions for three control embryos, recording the position of all seam cells three times per second from 530 mpf to hatch (>4.5 hours). These 150,000+ annotated image volumes (freely available on FigShare, https://figshare.com/s/6d498d74f98cca447d41) create a new resource for the systematic analysis of *C. elegans* embryo behavior.

## Assessing speed, direction of motion, and bout durations

Next, we used a variety of analytical approaches to quantify embryo motion. Mean squared displacement (MSD) is commonly used to describe the motion of particles and cells. Consistent with confined motion, MSD of embryonic seam cell nuclei saturated in minutes. Anterior seam cells had the highest MSD plateaus, which approximated the square of the egg's mean radius (*Figure 2a*). These results demonstrate that the embryo's anterior end explored the full eggshell volume. A 'diffusion coefficient' (D) for each seam cell nucleus was estimated by linear fitting over 10 seconds (i.e. 30 frames). This analysis revealed that anterior seam cell motility was greater than that exhibited by posterior cells (*Figure 2b*). We also observed a consistent developmental shift, with the least motility occurring at ≈600 mpf (*Figure 2c*). *Figure 2d* illustrates the developmental transitions of two seam cells. As predicted from their MSD (*Figure 2a*), H1 (an anterior cell) explores the full extent of the eggshell, while V6 (a posterior cell) is more confined. The H1 'diffusion coefficient' was virtually identical at 530 and 750 mpf; however, the later trajectory reveals a shift to more directed movement, indicating more mature embryonic behavior.

To further characterize embryo behavior, we considered speed and movements along the antero-posterior axis. For embryo speed, we measured displacement of left-right seam cell pair midpoints (H1 to V5) over 1 s (3 frames), and then calculated a weighted average based on the straight-line length of the body at that segment and volume (*Figure 2e*). Embryo speed varied with age, mirroring the developmental shift in seam cell diffusion coefficients described above (*Figure 2c and e*). A marked

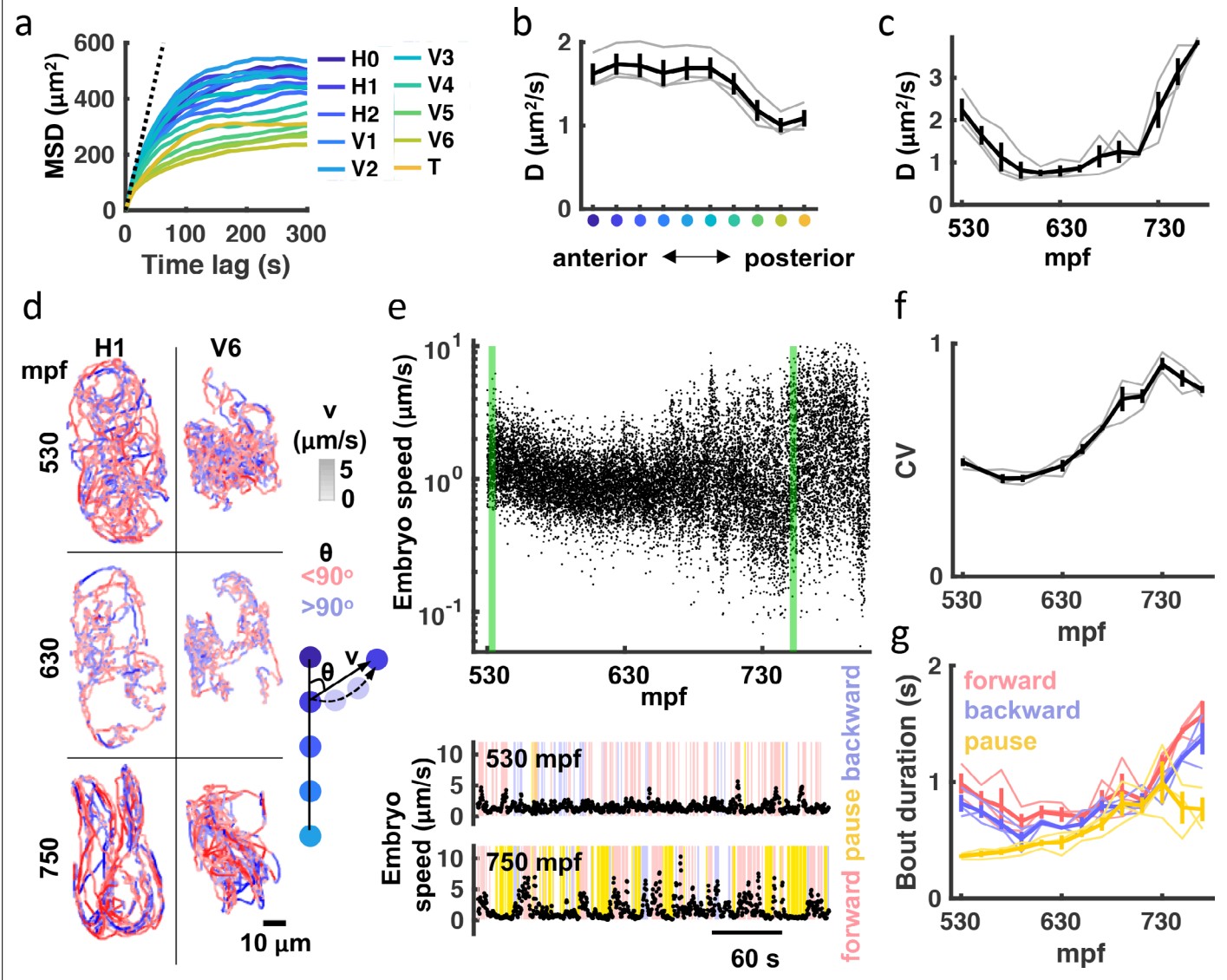

**Figure 2.** Transition to punctuated and directed movement ahead of hatching. (**a**) Mean square displacement (MSD) for each seam cell nucleus is plotted. Dotted line indicates predicted unconfined Brownian motion for the H1 seam cell. (**b**) Mean diffusion coefficient (**D**) over the entire recording is plotted for each seam cell pair. X-axis label colored as in (**a**). Individual embryo values (grey) and mean +/-SEM (black) are shown. (**c**) Diffusion coefficients for all seam cells were computed over 20 minute bins. Individual embryo values (grey) and mean +/-SEM (black) are shown. (**d**) 20 minute trajectories are shown for the H1 and V6 seam cell at the indicated ages (left). Cell velocity ('v') is indicated by the shade of tracks (see velocity LUT). Direction of motion along the anteroposterior axis (' $\theta$ ') is indicated by pink and blue. A schematic illustrating how 'v' and ' $\theta$ ' are defined is shown (right). (**e**) Embryo speed is plotted from 530 to 808 mpf. Forward, reverse, and pausing bouts are highlighted in 5-min embryo speed traces (bottom). Forward and backward movement were defined as $\bar{\theta}$ less than 45° and greater than 135°, respectively. Pauses were defined as speed< 0.5 µm/s. (**f**) Coefficient of variation in embryo speeds for 20-min bins are plotted. Individual embryo values (gray) and mean +/-SEM (black) are shown. (**g**) Mean duration of forward, backward, and pausing bouts in 20-min bins are plotted. Individual embryo values (faint traces) and mean +/-SEM (bold traces) are shown.

increase in the coefficient of variation of embryo speed highlights the extent to which movement becomes increasingly punctuated in the final hours of embryogenesis (*Figure 2e and f*), exhibiting both big displacements and periods of relative inactivity. Timing movements along the anteroposterior axis, we found that the duration of both forward and backward bouts increased over the final 2 hr of embryogenesis, as did the duration of pausing (*Figure 2e and g*). In summary, seam cell tracking reveals a consistent developmental progression in embryo behavior. Immature embryos move nearly continuously in short trajectories, while mature embryos show prolonged bouts of forward and backward movement punctuated with pausing.

## A high-throughput assay for assessing embryonic twitching

Seam cell tracking has two potential limitations as a strategy to analyze embryo behavior. First, the fluorescence imaging required for posture tracking could artificially distort the observed embryonic behaviors (e.g. due to subtle phototoxic effects). Second, although our imaging pipeline is semi-automated, seam cell tracking remains labor intensive. To address these concerns, we devised an independent high-throughput brightfield assay to assess overall embryonic motility. Brightfield images of up to 60 embryos were simultaneously acquired at 1 Hz, and frame-to-frame changes in pixel intensity were used as a proxy for embryo movement (*Figure 3a*; *Figure 3—video 1*). The resulting twitch profiles were highly stereotyped, exhibiting three salient phases: an immature active phase (≈450–550 mpf), followed by a relatively inactive period (≈550–650 mpf), followed by a second active phase (≥650 mpf; *Figure 3b*) with a broadened distribution of movement magnitudes (*Figure 3c*). We generated scalograms to visualize the temporal structure of twitch profiles over a range of timescales. This analysis revealed a prominent feature in the 20–40 mHz frequency band, occurring about an hour before hatching (*Figure 3d*). This feature was attributed to an increased propensity for prolonged pausing, as had been seen in seam cell tracks at this stage (*Figure 2g*). We call this behavioral signature slow wave twitch (SWT).

While the general pattern of behavioral maturation was highly stereotyped, absolute timings of the different phases varied between experiments. For example, the mean time from first twitch to hatch ranged from 4.5 to 6.5 hr (324.1+/-37.0 minutes, mean +/-SEM) across 31 brightfield recordings (*Figure 3—figure supplement 1*). This variability could result from differences in temperature or buffer salinity, both of which have been shown to influence the rate of development *Wood, 1988*; *Baris Atakan et al., 2020*. The poly-L-lysine used for sticking embryos to the coverslip is an additional potential source of variability. However, the timing of SWT scaled with hatch time (*Figure 3—figure supplement 1*), suggesting that the relative timing of development was consistent across experiments. In summary, similar motion profiles were observed in the brightfield and seam cell tracking assays, further suggesting that the observed progression of behaviors is an authentic feature of embryonic development.

## Eigen-embryos compactly describe posture and behavioral motifs

Adult worms crawling on a 2D agar surface explore a low dimensional postural space. A prior study showed that over 90% of variance in dorsoventral bending along the midline could be accounted for with just four principal components (PCs), which were termed 'eigenworms' *Stephens et al., 2008*. To describe embryo posture, we used seam cell positions to fit each side of the body with a natural cubic spline and then computed dorsoventral bending angles between adjacent seam cells (totalling 18 bend angles in each volume; *Figure 4a and b*). Four PCs captured approximately 88% of variation in the 18 angles (*Figure 4c and d*). The corresponding eigenvectors of the four leading components (termed eigen-embryos) were stereotyped between animals. For example, PC1 captures ventral or dorsal coiling (i.e. all ventral or all dorsal body bends, respectively) while PC2 describes postures with opposing anterior and posterior bends (*Figure 4e*). In this framework, embryonic posture can be approximated by a linear combination of eigen-embryos. The contribution of eigen-embryos shifted consistently across development, with less variance accounted for by PC1 as PC2 and PC3 gained prominence (*Figure 4d*). PC2 and PC3 approximate sinusoids with a phase difference of about 90° that can be combined to generate travelling waves of dorsoventral bending. The developmental shift towards PC2 and PC3 more closely approximates the adult motion pattern, where the top two PCs also describe sinusoids with a 90° phase shift *Stephens et al., 2008*. The relatively limited dimensionality of dorsoventral bending observed in both adults and embryos is likely a consequence of muscle anatomy, with electrically coupled muscle bundles running ventrally and dorsally along the body *White et al., 1986*.

Because seam cells are bilaterally symmetric, we could not assign left versus right identity and consequently could not distinguish dorsal from ventral bends. At hatching, dorsal and ventral muscles are controlled by distinct classes of motor neurons *White et al., 1978*; *Hallam and Jin, 1998*; *Walthall et al., 1993*. We wondered if this difference in connectivity results in a systematic asymmetry in embryo postures. Consistent with this idea, we observed an asymmetry in the PC1 amplitude distribution of late-stage embryos (*Figure 4—figure supplement 1a*). To determine if the bias was consistently dorsal or ventral, we acquired a supplemental dataset of 16 embryos expressing additional

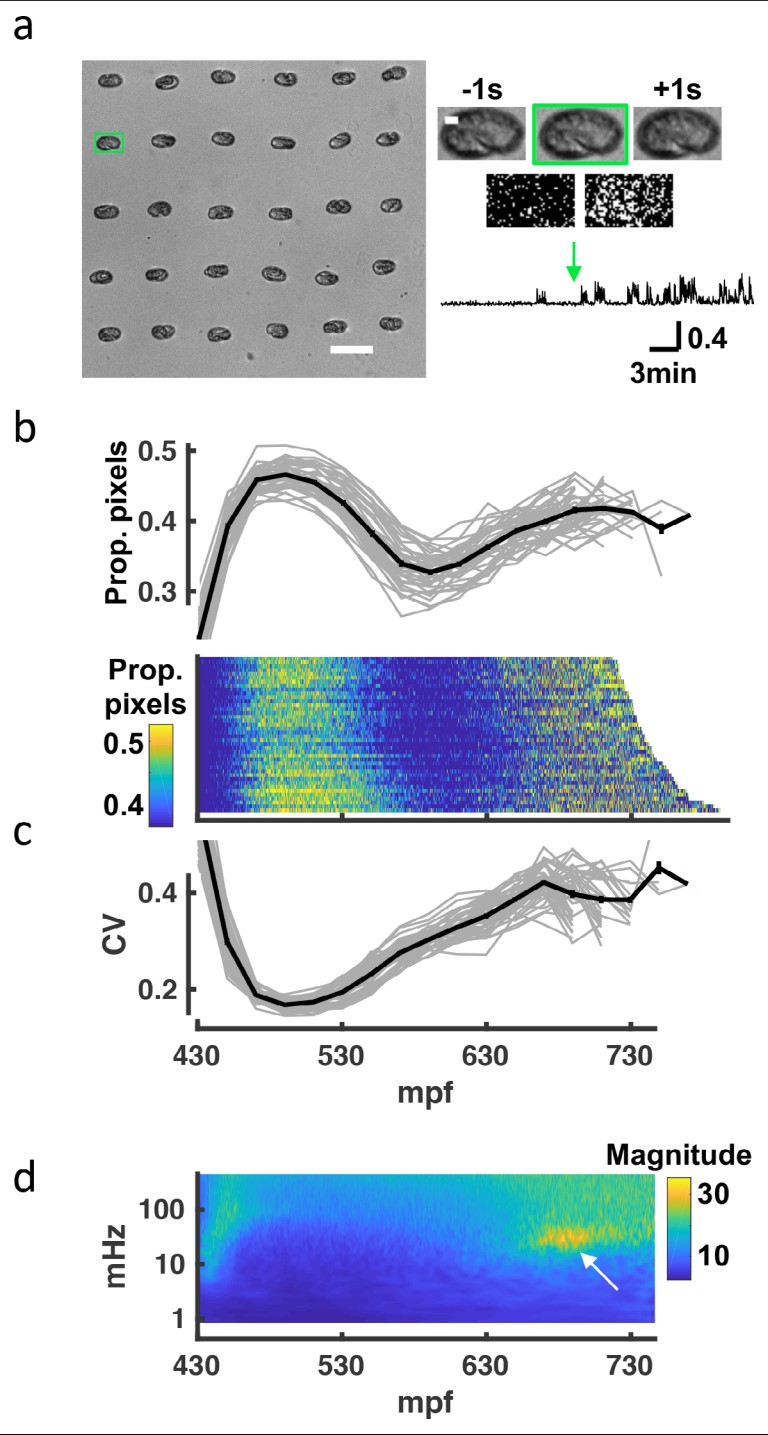

**Figure 3.** A brightfield motility assay reveals a stereotyped pattern of embryonic behavior maturation. To independently assess embryo behavior, we developed a brightfield motility assay. Motion of each embryo is estimated by the fraction of pixels changing intensity between frames. The 'twitch profiles' only report crude motion and are unable to distinguish more detailed aspects of behavior (e.g., direction or posture). (**a**) Embryos arrayed for a brightfield assay are shown (left). Scale bar, 100 μm. Higher magnification views of the highlighted embryo (scale bar, 10μm) over three consecutive frames and frame subtraction images are shown (right). A representative twitch trace illustrating proportion of pixels changing at twitch onset (≈430 mpf) is shown (bottom). *Figure 3—video 1* shows a representative brightfield recording. (**b**) Top: Average pixel changes aligned at twitch onset are plotted (in 20-min bins). Individual embryo values (gray) and mean +/-SEM (black) are shown for n=40 embryos. Bottom: Brightfield twitch profile, with each line corresponding to a single embryo. Proportion of pixels

*Figure 3 continued on next page*

*Figure 3 continued*

changing intensity (smoothed over 10 s) is indicated by the color LUT. (**c**) Coefficient of variation of pixel changes aligned at twitch onset (in 20-min bins) are plotted. Individual embryo values (gray) and mean +/-SEM (black) are shown. (**d**) Mean twitching scalogram is shown. The signal in the 20–40 mHz frequency band around 670–710 mpf (arrow) represents the SWT behavior.

The online version of this article includes the following video and figure supplement(s) for figure 3:

**Figure supplement 1.** Time from twitch to SWT versus twitch to hatch for 31 brightfield recordings of wild-type embryos.

**Figure 3—video 1.** An embryo array for the brightfield assay.

https://elifesciences.org/articles/76836/figures#fig3video1

fluorescent markers. The added markers allowed us to unambiguously identify left and right seam cell nuclei by manual inspection. This supplemental dataset was collected at lower temporal resolution (5 min intervals), thereby precluding detailed behavioral tracking. The postures exhibited by embryos in the supplemental dataset were described by a similar set of eigen-embryos as the original dataset (*Figure 4—figure supplement 1c*). In the final hours of embryogenesis, the postures in the supplemental dataset exhibited a dorsal bias (*Figure 4—figure supplement 1b*). By assuming a dorsal bias for the original dataset, we were able to distinguish left and right seam cells (and consequently dorsal and ventral body bends) in the three embryos recorded at 3 Hz.

Our analysis of embryo postures is consistent with a prior study of embryo behavior *Hall and Hedgecock, 1991*. Using only differential interference contrast (DIC) microscopy and manual visual inspection, this study describes embryo motion as flips about the anteroposterior axis that decline in frequency ahead of the emergence of the 'larval locomotor pattern'. This proposed developmental progression was readily apparent in posture space. Flips about the anteroposterior axis were seen as oscillating PC1 amplitudes, which signifies alternating dorsal and ventral coiling (*Figure 4f*; *Figure 4— figure supplement 2a and b*). The 'larval locomotor pattern' was best captured by PC2 and PC3, for which phase-shifted cycling describes dorsoventral bend propagation along the body axis (*Figure 4f*; *Figure 4—figure supplement 2c*), reminiscent of sinuous crawling in larvae and adults. Thus, the posture library captures a developmental progression from immature (dorsoventral flipping) to mature (sinusoidal crawling) embryo behavior.

## Synaptic signaling is required for mature motion, but not dorsoventral flipping

The behavior of late-stage embryos exhibits several potential signatures of neuronal control (increasingly directed movement, dorsal coiling bias, and sinuous crawling). To establish a role for synaptic signaling, we analyzed *unc-13(s69)* mutants which have a nearly complete block in synaptic vesicle fusion and (consequently) profound movement defects *Richmond and Jorgensen, 1999*. In late-stage embryos (750 mpf), *unc-13* mutant movement was strongly impaired, as indicated by shorter seam cell trajectories and smaller diffusion coefficients (*Figure 5a*). Although their motion was severely restricted, *unc-13* mutants continued subtle movements in place, suggesting that spontaneous muscle contractions persist even when synaptic transmission is blocked. By contrast, at 530 mpf seam cell diffusion coefficients for *unc-13* mutants were indistinguishable from controls (*Figure 5b*), implying that synaptic transmission is not required for the behavior of immature embryos. To further investigate the role of synaptic transmission in immature behavior, we analyzed flipping, which is the most salient behavior at this stage. For this analysis, flips were defined as transitions between dorsal and ventral coiling, i.e. from fully dorsally to fully ventrally bent postures and vice versa. At 530 mpf, control and *unc-13* mutants flipped 3–4 times per minute, with 75% of the transitions executed in 10 s (*Figure 5c*). Flips can be considered translations along the PC1 axis (*Figure 4f*, *Figure 4—figure supplement 2a and b*). We used mean PC2 and PC3 amplitudes during the transition to define 4 flip motifs: PC2-, PC2+, PC3-, and PC3+ (*Figure 5d*). We found that trajectories through posture space were indistinguishable in control and *unc-13* mutants. For example, in both genotypes, ventral to dorsal flips usually started in the tail, with intermediate postures having a posterior dorsal bend (i.e. the PC2- motif; *Figure 5d*). *Figure 5e* summarizes the prevalence of each flip motif. Note that dorsal to ventral flips were also more likely to initiate in the tail (i.e. the PC2 +motif; *Figure 5e*). By

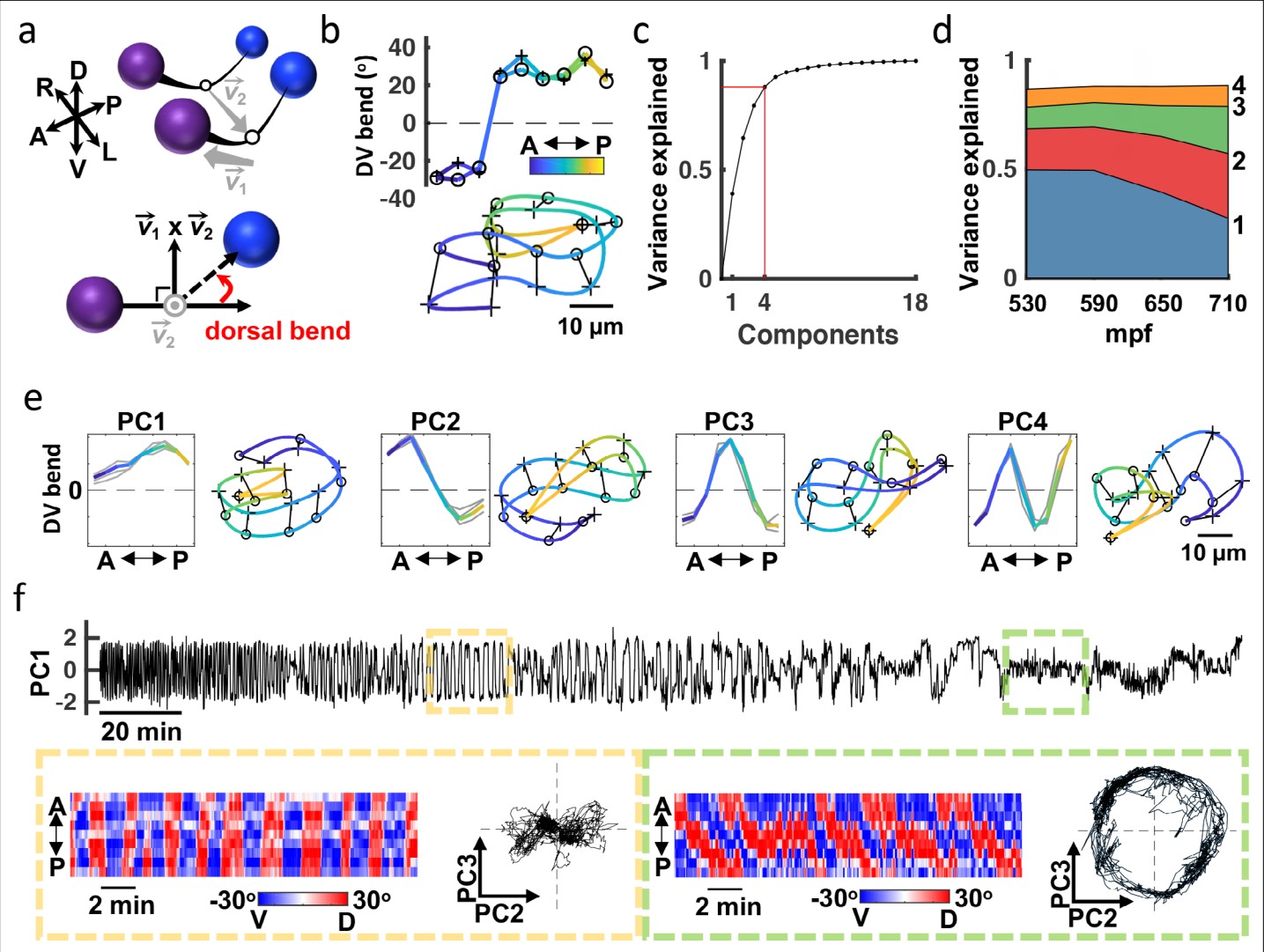

**Figure 4.** Eigen-embryos compactly describe behavioral maturation. (**a**) A schematic illustrating how dorsoventral (DV) bends are defined. Top: Seam cell nuclei on each side of the body are fit with a natural cubic spline (black line). Vector $\vec{v_2}$ links the midpoints between adjacent seam cell nuclei (open circles). Bottom: View looking down $\vec{v_2}$ 2 to highlight a DV bend angle on one side of the body (red arrow). (**b**) DV bend angles between all adjacent seam cells are used to define an embryo's posture. DV bends along the left (+) and right (o) sides of an embryo (top) are plotted (top). The posture model for this embryo is shown (bottom). Position along the anteroposterior axis is indicated by the color gradient. (**c**) The fraction of the total variance captured by reconstructing postures using 1 through all 18 principal components is plotted. (**d**) Fraction of the total variance captured by the first 4 principal components is plotted as a function of embryo age (mpf). (**e**) Eigen-embryos (i.e. eigenvectors for the first 4 PCs) derived from the three individual embryos in grey and from pooled datasets in color (left). Sample postures drawn from the 95th percentile of amplitudes are shown for the indicated PCs (right). See *Figure 4—figure supplement 1* for supplemental embryos. (**f**) A representative PC1 amplitude trace is shown for an embryo from 530 mpf until hatch (top). DV bend profiles in two 20 min windows are compared (bottom). At 610 mpf (yellow), motion is dominated by flipping between all dorsal and all ventral body bends. At 750 mpf (green), motion is dominated by dorsoventral bend propagation along the A/P axis. The corresponding PC2 and PC3 amplitudes for these time windows are shown. At 750 mpf, travelling waves are captured by phase-shifted cycling of PC2 and PC3. See *Figure 4—figure supplement 2* for more embryos.

The online version of this article includes the following figure supplement(s) for figure 4:

**Figure supplement 1.** Eigen-embryos derived from a supplemental dataset of 16 animals.

**Figure supplement 2.** PC amplitudes and DV bends for 3 control embryos.

contrast, locomotion in larvae (and older embryos; *Figure 4—figure supplement 2c*) is dominated by bends propagating from head to tail, likely requiring neuronal coordination. These results suggest that motion in immature embryos is largely independent of UNC-13-mediated synaptic transmission, although an immature form of synaptic transmission (not requiring UNC-13) cannot be ruled out.

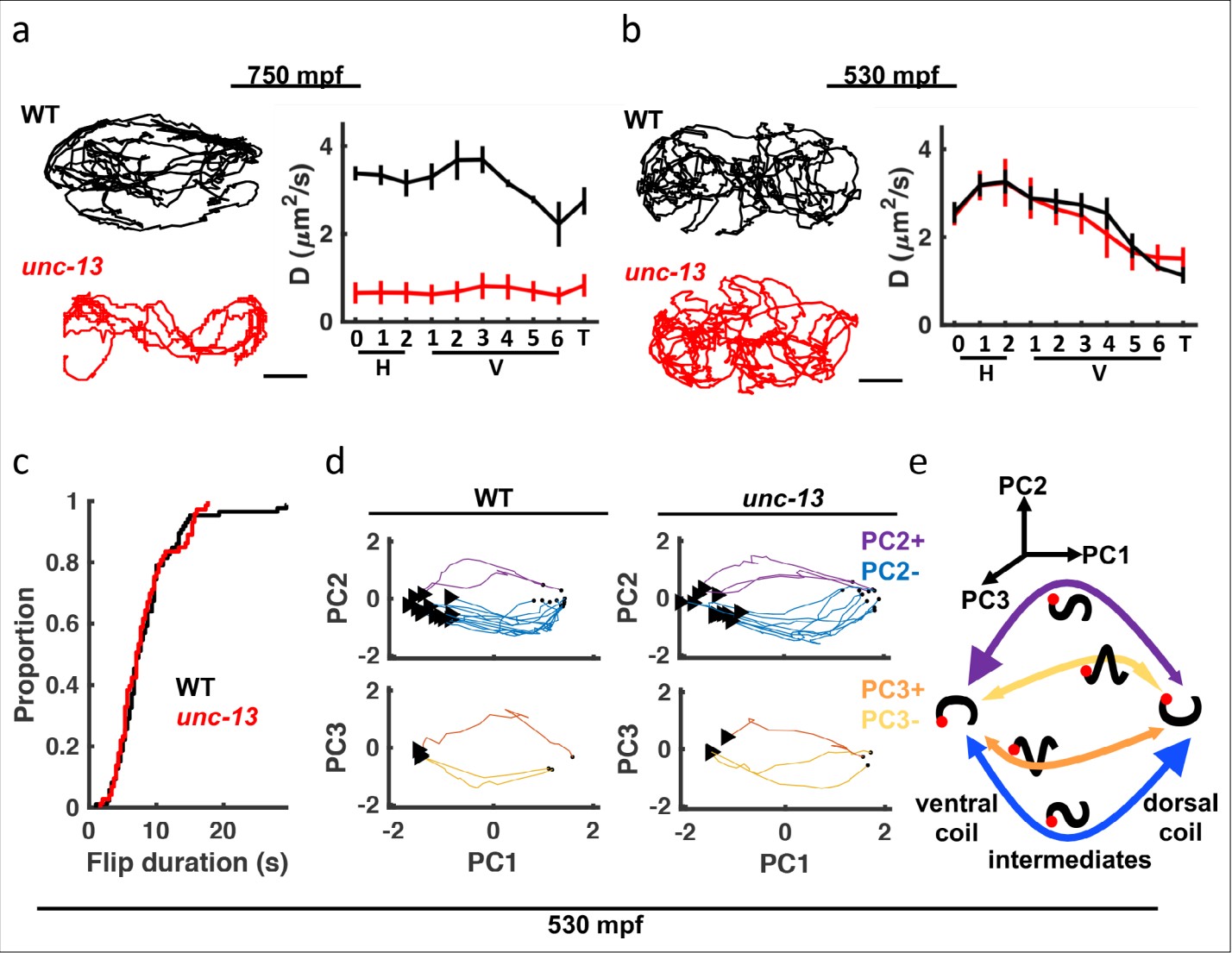

**Figure 5.** *unc-13 mutants have a late-stage motility defect.* (**a–b**) Seam cell motions are compared in WT and *unc-13* mutant embryos at 750 mpf (**a**) and 530 mpf (**b**). Representative 10 min trajectories for the H1 seam cell (left) and mean diffusion coefficients for all seam cell pairs (right; mean +/-SEM) are shown. Scale bar, 10 $\mu m$. Sample sizes, 750 mpf (3 WT, 3 *unc-13*); 530 mpf (3 WT, 2 *unc-13*). (**c**) Cumulative distribution functions are shown for duration of transitions between dorsal and ventral coils (i.e. flips) from 530 to 540 mpf. Flip durations observed in WT and *unc-13* mutants cannot be distinguished. (**d**) Posture transitions during flipping were unaltered in *unc-13* mutants. An embryo's transitions through posture space during flipping is compared in WT and *unc-13* mutants (530–540 mpf; only transitions executed in ≤10 s are plotted). Flips were categorized into four motifs based on the mean amplitude of PC2 and PC3 during the transition (PC2+, PC2-, PC3+, and PC3-). The start (arrowhead) and end (dot) of each flip is indicated. (**e**) Summary of coiling transition trajectories in posture space at 530–540 mpf, with arrowhead sizes indicating the frequency of occurrence. Color-code same as in (**d**). Schematized worms (black squiggles) are shown with ventral down and head (red dot) to the left.

## Slow wave twitch (SWT): A rhythmic behavior dependent on the nervous system

To accelerate mutant analysis, we turned to the brightfield assay (*Figure 3*). The profound motility defect observed in late-stage *unc-13* mutant embryos by seam cell tracking (*Figure 5a*) was not immediately obvious in the brightfield assay (*Figure 6a*), most likely due to the strong contribution of spontaneous muscle contractions to pixel intensity changes. In search of a synaptic influence on movement, we analyzed embryo twitching in the frequency domain. Isolating the *unc-13*-dependent signal from the wild-type scalogram revealed the 20–40 mHz SWT feature described above (*Figure 6b*). To quantify SWT, we scanned the twitch profile of each embryo for the 15 min in which 20–40 mHz most dominated the power spectrum. The SWT power ratio refers to the proportion of the power

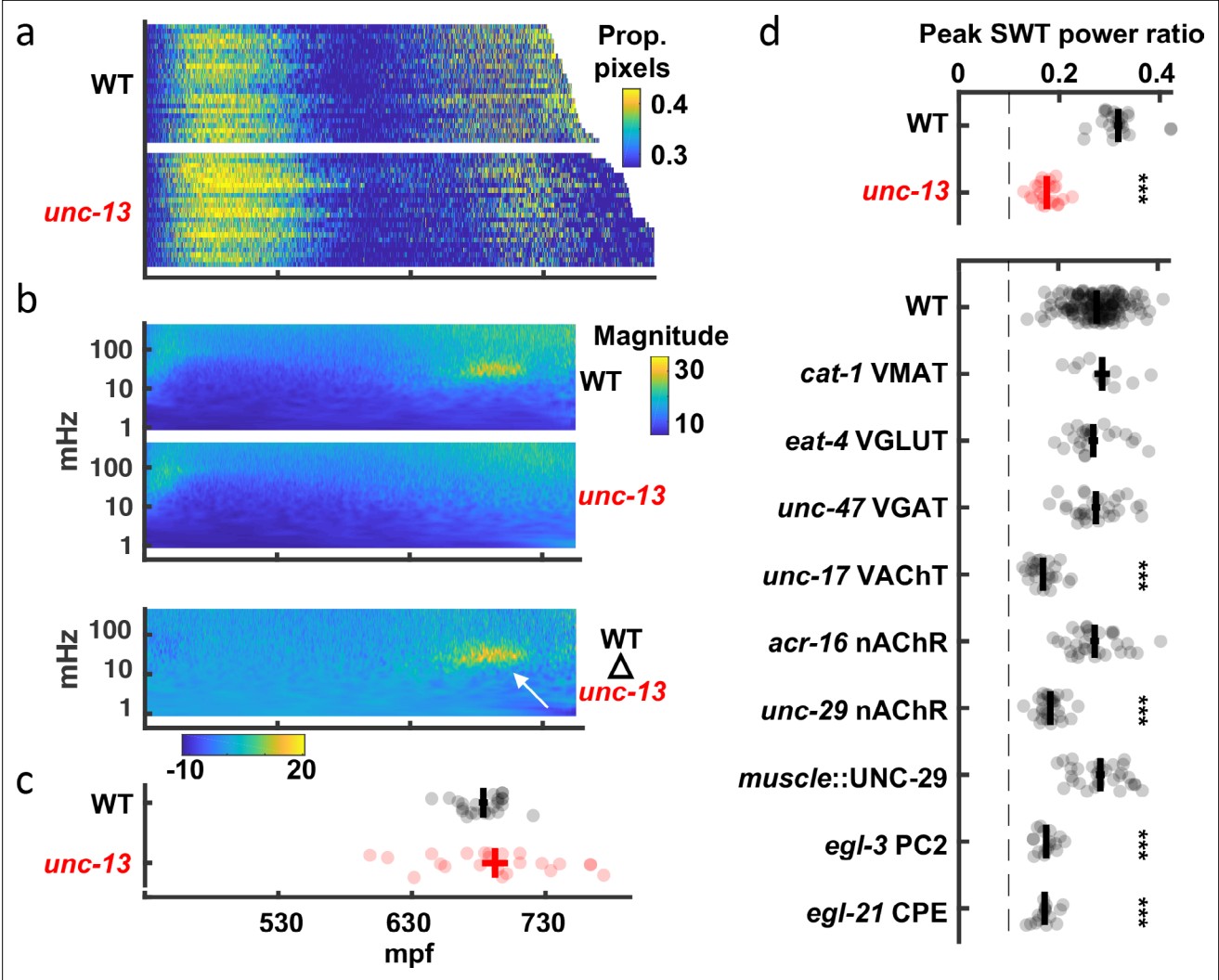

**Figure 6.** Slow wave twitch (SWT) requires cholinergic and neuropeptide signaling. (**a**) Brightfield twitch profiles are shown for wild-type (WT; n=24) and *unc-13* mutants (n=23). Each line corresponds to a single embryo. All embryos are aligned at twitch onset (430 mpf). Proportion of pixels changing intensity (smoothed over 10 s) is indicated by the color LUT. (**b**) SWT behavior is absent in *unc-13* mutants. Mean twitching scalograms for wild-type and *unc-13* mutants and the difference scalogram (WT – *unc-13*) are shown. The signal in the 20–40 mHz frequency band around 660–720 mpf (arrow) is SWT behavior. (**c**) The timing of peak relative power in the SWT frequency band (20–40 mHz) is plotted (mean +/-SEM, with each dot corresponding to an embryo). (**d**) SWT behavior requires cholinergic and neuropeptide signaling. Peak SWT power ratios compared for several synaptic signaling mutants. Mean +/-SEM are shown, with each dot corresponding to an embryo. Sample sizes are as follows: WT, n=156; *cat-1* VMAT, n=11; *eat-4* VGLUT, n=24; *unc-47* VGAT, n=30; *unc-17* VAChT, n=26; *acr-16* nAChR, n=31; *unc-29* nAChR, n=29; muscle::UNC-29, n=31; *egl-3* PC2, n=17; *egl-21* CPE, n=16. One-way ANOVA and Tukey's honestly significant difference criterion were used to compare strains. Significant differences from wild-type are indicated (***, $p < 0.001$). The dashed line indicates the peak SWT power ratio derived from shuffled wild-type data. *Figure 6—figure supplement 1* shows the full twitch profiles for strains with SWT defects.

The online version of this article includes the following figure supplement(s) for figure 6:

**Figure supplement 1.** Full twitch profile for SWT mutants.

spectrum accounted for by the 20–40 mHz frequency band. In wild-type animals, the SWT power ratio consistently peaked about an hour before hatching (*Figure 6c*). As suggested by the scalograms, peak SWT power ratios were markedly diminished in *unc-13* mutants compared to wild-type (*Figure 6d*). Thus, SWT represents a rhythmic embryonic behavior with a period of 25–50 s that requires synaptic transmission.

To identify neurons required for SWT, we analyzed mutants disrupting specific neurotransmitters (*Figure 6d*, *Figure 6—figure supplement 1*). Peak SWT power ratios were unaffected by loss of the vesicular transporters for biogenic amines (*cat-1* VMAT), glutamate (*eat-4* VGLUT), or GABA (*unc-47*

VGAT). By contrast, peak SWT power ratios were strongly diminished in mutants lacking the vesicular transporter for acetylcholine (*unc-17* VAChT; *Figure 6d*), indicating that cholinergic transmission plays a prominent role in generating SWT behavior. Although motility was generally decreased in *unc-17* mutants (*Figure 6—figure supplement 1*), changes in the magnitude of motion alone cannot explain changes in the relative power of the SWT frequency band. The SWT power ratio quantifies the temporal structure of motion. Body wall muscles express two classes of nicotinic acetylcholine receptors: homopentameric ACR-16 receptors and heteropentameric levamisole receptors containing UNC-29 subunits *Richmond and Jorgensen, 1999*; *Touroutine et al., 2005*. Mutants lacking UNC-29 had significantly reduced peak power ratios in the SWT frequency band and this defect was rescued by a transgene expressing UNC-29 in muscles (*Figure 6d*). By contrast, peak SWT power ratios were unaffected in *acr-16* mutants (*Figure 6d*). Thus, SWT requires cholinergic signaling through UNC-29-containing receptors in muscles. Peak SWT power ratios were also strongly diminished in mutants lacking two pro-neuropeptide processing enzymes (*egl-3* PC2 and *egl-21* CPE; *Figure 6d*), implicating neuropeptide signaling in SWT behavior.

## RIS release of FLP-11 neuropeptides and GABA promote SWT quiescence

Peak SWT behavior coincides with a period of increased pausing, as detected by seam cell tracking (*Figure 2e and g*); consequently, we hypothesized that signal in the 20–40 mHz frequency band was a consequence of brief quiescent bouts. Two neurons (RIS and ALA) are known to promote behavioral quiescence in larvae and adults *Turek et al., 2016*; *Turek et al., 2013*; *Van Buskirk and Sternberg,*

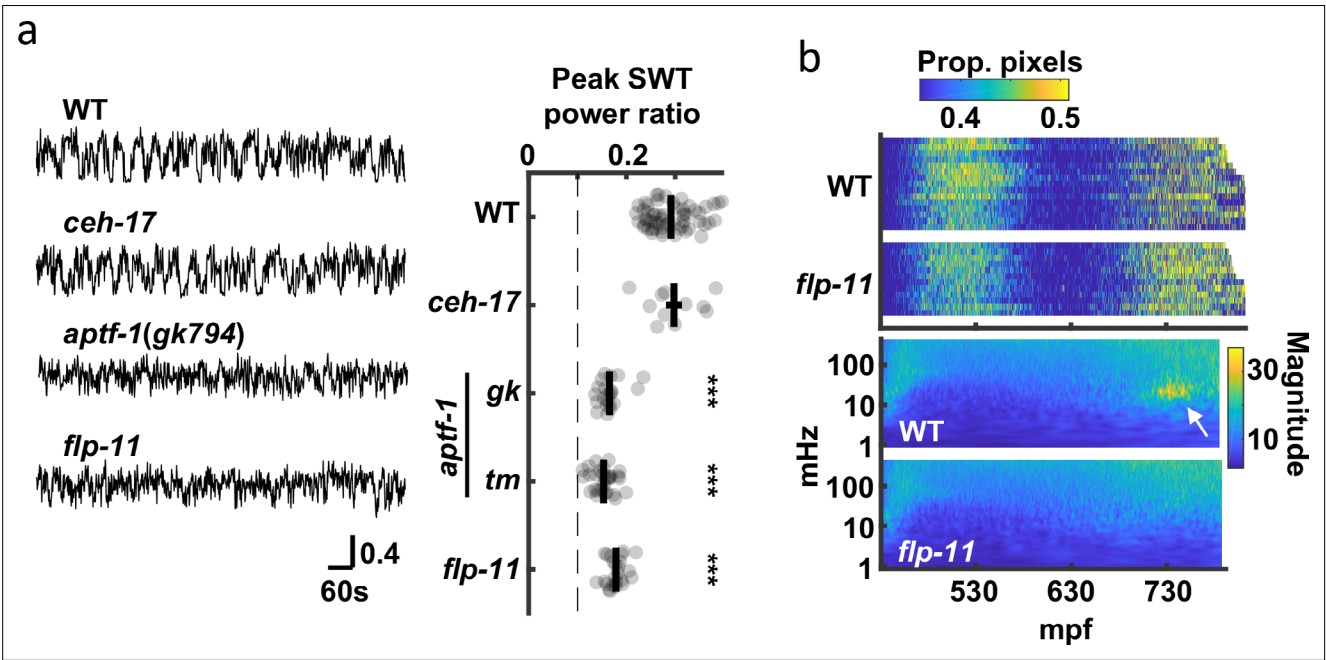

**Figure 7.** SWT behavior requires function of the RIS neuron. (**a**) The contribution of two quiescence promoting neurons (RIS and ALA) to SWT behavior is evaluated. SWT behavior is deficient in mutants lacking RIS function (*aptf-1* and *flp-11*) but is unaffected in those lacking ALA function (*ceh-17*). Left: Representative twitch traces at peak SWT power ratio are shown. Amplitudes indicate the proportion of pixels changing intensity. Right: Peak relative power in the SWT frequency band (20–40 mHz) is plotted for the indicated genotypes. Mean +/-SEM, with each dot corresponding to an embryo. The dashed line denotes peak SWT power ratio derived from shuffled data. Two *aptf-1* alleles (*gk794* and *tm3287*) were analyzed. One-way ANOVA and Tukey's honestly significant difference criterion were used to compare strains. Significant differences from wild-type are indicated (***, $p < 0.001$). (**b**) Top: Brightfield twitch profiles are shown for wild-type (WT) and *flp-11* mutants. Each line corresponds to an individual embryo aligned at twitch onset (430 mpf). The proportion of pixels changing intensity (smoothed over 10 seconds) is indicated by the color LUT. See *Figure 7—figure supplement 1* for cumulative distribution functions. Bottom: Wild-type and *flp-11* scalograms. Note the SWT signature in WT (arrow) is missing in *flp-11* mutants.

The online version of this article includes the following figure supplement(s) for figure 7:

**Figure supplement 1.** Cumulative probability histograms of embryo motion (assessed by proportion of pixels changing intensity) are plotted for the indicated genotypes.

**Video 1.** Seam cell nuclei fit with a natural cubic spline from H0 (blue) to tail (yellow; left). Scale bar, 10 $\mu m$. Projection of postures onto three planes in PC space, with red x's denoting pauses (right). Original data acquired at 3 Hz.

https://elifesciences.org/articles/76836/figures#video1

*2007*; *Nath et al., 2016*. Prompted by these results, we asked if SWT behavior is disrupted in mutants lacking RIS (*aptf-1* AP-2) or ALA (*ceh-17* PHOX2A) function *Turek et al., 2013*; *Pujol et al., 2000*; *Van Buskirk and Sternberg, 2007*. Consistent with a role for RIS, we found that peak SWT power ratios were markedly reduced in *aptf-1* mutants but were unaffected in *ceh-17* mutants (*Figure 7a*). RIS neurons elicit larval quiescence by secreting neuropeptides encoded by the *flp-11* gene *Turek et al., 2016*. We found that *flp-11* mutants exhibited a superficially normal pattern of embryo motion (i.e. early and late active phases with an intervening period of decreased mobility), suggesting that the overall developmental progression of embryonic behavior was unaffected (*Figure 7b*, *Figure 7—figure supplement 1*). By contrast, SWT behavior was dramatically reduced in *flp-11* mutants, as indicated by the absence of the 20–40 mHz feature in scalograms (*Figure 7b*) and by a significantly reduced peak SWT power ratio (*Figure 7a*). These data suggest that late-stage movement is structured by periodic release of FLP-11 from RIS, which results in increased power in the 20–40 mHz frequency band.

To further describe SWT behavior, we analyzed embryo postures at the developmental stage corresponding to peak SWT behavior. We found that pauses occurred throughout posture space (*Video 1*), suggesting that SWT quiescent bouts are not associated with specific body postures.

To directly relate RIS activity to embryo behavior, we simultaneously monitored motion and activity of the RIS cell body with a genetically encoded calcium sensor, GCaMP6f *Chen et al., 2013*, fused to a calcium insensitive fluorophore, mCherry. Tracking GCaMP/mCherry ratios in the RIS cell body in embryos aged at least 645 mpf (*Figure 8a and b*), we observed RIS calcium transients at a rate within the SWT frequency band (24.8+/-1.6 mHz, mean +/-SEM; *Figure 8c*). By contrast, ALA neuron calcium transients occurred far less frequently (1.9+/-0.3 mHz, mean +/-SEM). In mutants lacking FLP-11 neuropeptides, the rate and duration of RIS transients were unaltered (*Figure 8c and d*); however, transient amplitudes were significantly increased (*Figure 8e*). Larger calcium transient amplitudes in *flp-11* mutants suggests that FLP-11 may inhibit RIS activity in an autocrine manner. Behavioral quiescence was substantially reduced but not eliminated in *flp-11* mutants (*Figure 8c and d*). Using the onset of calcium transients to align behavior and fluorophore intensities (GCaMP and mCherry), we found that the GCaMP signal (and not the mCherry signal) was negatively correlated with head speed (*Figure 8f*), confirming that RIS activation was associated with behavioral slowing. This relationship persisted in *flp-11* mutants, however compared to wild-type, the behavioral slowdown was more transient (*Figure 8f*) and less likely to lead to a pause (*Figure 8g*). The residual behavioral slowing found in *flp-11* mutants could be mediated by another RIS neurotransmitter, for example, GABA. Consistent with this idea, compared to *flp-11* single mutants, the behavioral slowdown and quiescence associated with RIS calcium transients was diminished in *flp-11;unc-25* GAD double mutants, which are deficient for GABA synthesis (*Figure 8—figure supplement 1*). These results suggest that a prominent feature of late-stage embryonic behavior is a rhythmic pattern of quiescence elicited by two RIS neurotransmitters, FLP-11 and GABA. GABA release was associated with transient behavioral slowing, whereas FLP-11 release was essential for sustained pausing.

## Sensory response during SWT is dampened by FLP-11

Other forms of quiescence are associated with diminished responsiveness to arousing stimuli. To see if SWT shares this property, we analyzed the behavioral response to light. In adults and larvae, short-wavelength light has been shown to elicit arousal via endogenous photoreceptor LITE-1 *Edwards et al., 2008*; *Liu et al., 2010*; *Gong et al., 2016*. Here, we monitored motion and activity of the RIS cell body following a UVB light pulse (285 nm). In these experiments, a 10-s irradiance was delivered during peak SWT (1 hr before hatching). Stimulus trials were retrospectively identified as those occurring when RIS was active (RIS_on) or inactive (RIS_off). UVB elicited behavioral arousal in embryos

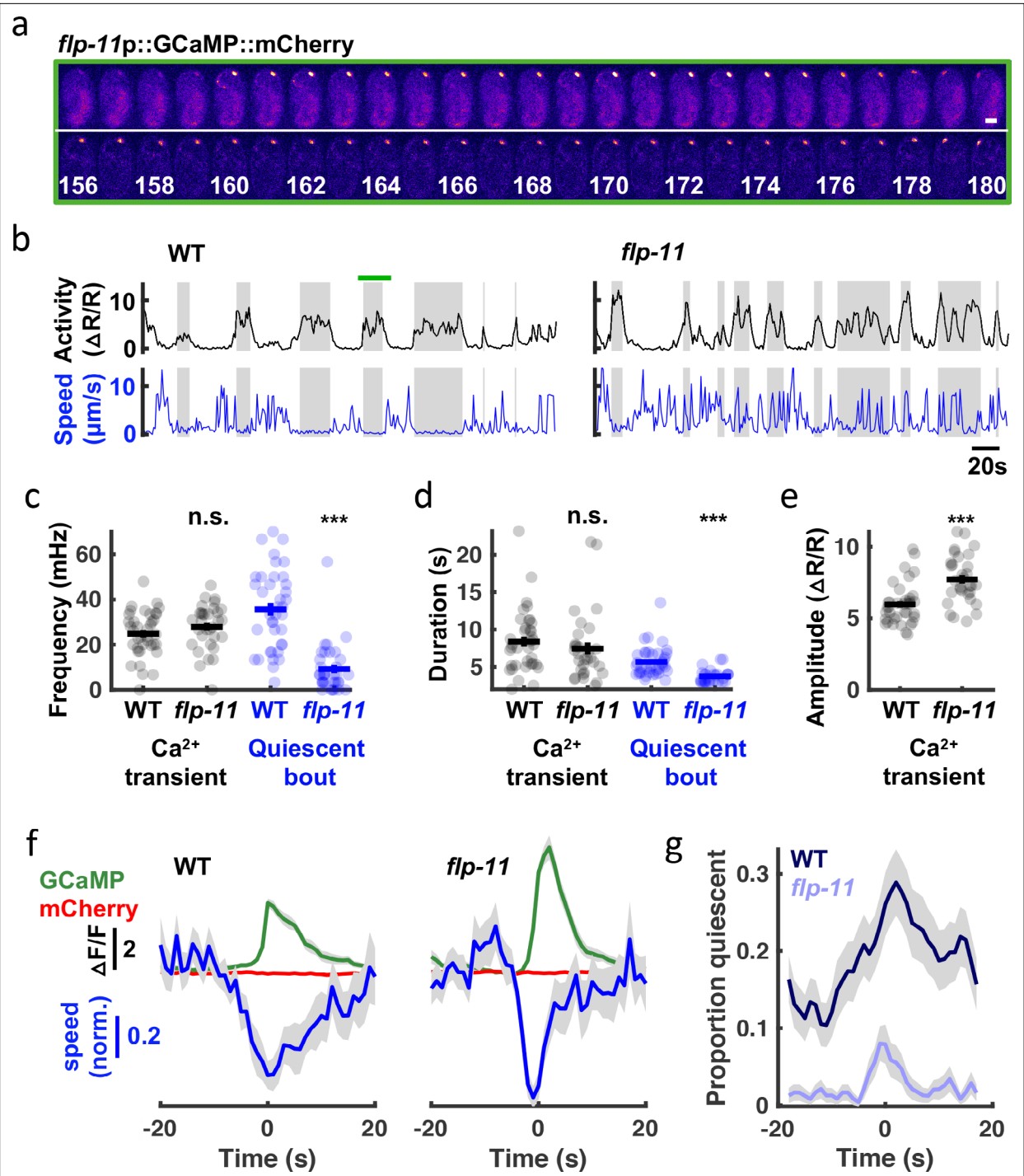

**Figure 8.** SWT behavior correlates with rhythmic RIS activity. (**a**) Time lapse images of a WT embryo expressing GCaMP (top) fused to mCherry (bottom) in RIS. A single confocal plane is shown. Time stamp, s. Scale bar, 10 μm (**b-g**) RIS activity (assessed by cell body GCaMP) and embryo motion (assessed by RIS cell body displacement) were simultaneously recorded in individual embryos. (**b**) Representative activity (top) and motion (bottom) traces are shown for WT and *flp-11* mutants. Gray bands indicate calcium transient durations (defined by widths at half peak amplitude). The green bar spans the period shown in (**a**). Calcium transient and quiescent bout frequency (**c**) and durations (**d**), and calcium transient amplitudes (**e**) are plotted. Mean +/- SEM are indicated. Each dot represents an individual embryo. Significant differences from WT (***, *p* < 0.001) were determined by an unpaired two-sample t-test. (**f**) Traces of median normalized speed and GCaMP and mCherry intensity are aligned to calcium transient onset (Time = 0 s defined as the time of half peak GCaMP/mCherry ratio). Behavioral slowing was more transient in *flp-11* mutants. Behavioral slowing was not correlated with mCherry intensity, suggesting that motion artifacts cannot explain the correlation with GCaMP intensity. (**g**) Quiescent fraction traces aligned to RIS calcium transient onset are plotted. Quiescence was defined as RIS displacement less than or equal to 0.5 μm per second for at least 3 s. In panels f and

*Figure 8 continued*

g, colored lines and grey shading correspond to mean +/- SD of the sampling distribution estimated by hierarchical bootstrapping for n=269 (WT) and n=261 (*flp-11*) calcium transients from n=38 and n=32 embryos, respectively. See *Figure 8—figure supplement 1* for *unc-25* GAD mutant data.

The online version of this article includes the following figure supplement(s) for figure 8:

**Figure supplement 1.** GABA and FLP-11 promote behavioural slowing and pausing.

(*Figure 9a and b*). Rapid and robust arousal was apparent in both RIS$_{off}$ and RIS$_{on}$ trials (*Figure 9a*), confirming that SWT quiescence was reversible. The duration of UVB evoked behavioral arousal was increased in *flp-11* mutants compared to control embryos (*Figure 9a–d*; compare the post-stim speed in panels *Figure 9b* versus *Figure 9d*). Together, these results suggest that FLP-11 dampens responses to UVB during the peak SWT period. Coincident with behavioral arousal, UVB irradiance evoked an acute inhibition of RIS activity (*Figure 9a–d*), as previously shown with blue light in larvae *Wu et al., 2018*. Given the negative correlation between RIS activation and head speed (*Figure 8f*), inhibition of RIS likely contributes to the behavioral arousal elicited by UVB. Both UVB evoked behavioral arousal and RIS inhibition were eliminated in mutants lacking the LITE-1 photoreceptor (*Figure 9e and f*).

To further investigate the impact of SWT on sensory responses, we used the brightfield twitch assay to monitor behavioral responses to longer wavelength visible light. Similar to UVB, visible light evoked an arousal response that was prolonged in *flp-11* mutants (*Figure 9—figure supplement 1*). Thus, arousal responses to aversive light stimuli are exaggerated in mutants lacking SWT quiescence. Unlike UVB, arousal elicited by visible light was followed by motion slightly (albeit significantly) lower than baseline (*Figure 9—figure supplement 1*). This inhibited motion was lost in *flp-11* mutants and may represent a homeostatic response to the arousing stimulus. Collectively, these results suggest that SWT quiescence is reversible and that behavioral responses to arousing stimuli are dampened by FLP-11 during SWT.

## Discussion

Here, we describe a systematic analysis of behavioral maturation in *C. elegans* embryos. Our analysis leads to three primary conclusions. First, embryos follow a stereotyped behavioral trajectory: early flips between dorsal and ventral coiling (most often initiated in the tail) slowing into a period of reduced motility, followed by emergence of mature behaviors (including sinuous crawling, directed motion, and rhythmic quiescent bouts; *Figure 10*). Second, synaptic transmission is required for late-stage movement, but not for early dorsoventral flipping nor the intervening period of reduced motility. Third, the rhythmic bouts of behavioral quiescence punctuating late-stage movement are elicited by rhythmic RIS activation and secretion of two RIS transmitters, FLP-11 and GABA. Below we discuss the significance of these findings.

### Three phases of embryonic behavior

Our results suggest that embryos exhibit a stereotyped program for behavioral maturation in the final few hours before hatching, which comprises at least three phases (early flipping, an intermediate phase of reduced motility, and a late phase of mature motion). This general progression was apparent by both seam cell tracking and by brightfield motility assays. It is likely that further analysis of posture libraries will identify additional embryonic behaviors.

Following elongation, embryo behavior was initially dominated by flipping between all dorsal and all ventral body bends (*Figure 4—figure supplement 2a and b*). Flipping behavior was not disrupted in *unc-13* mutants, implying that it does not depend on synaptic transmission (*Figure 5c and d*). Flipping could be mediated by intrinsic oscillatory activity in muscles or by a form of neuronal signaling that persists in the absence of UNC-13 (e.g., gap junctions or an unconventional form of synaptic vesicle exocytosis). Because flipping comprises alternating all dorsal and all ventral bends, there must be some mechanism to produce anti-correlated ventral and dorsal muscle contractions. It will be interesting to determine what drives immature flipping and whether this early behavior is required in some way for the subsequent emergence of mature behaviors.

Early flipping behavior is followed by a period of decreased motion. This slowdown is apparently not neuronally evoked, as it requires neither neuropeptide processing enzymes nor UNC-13 (*Figure 6a*, *Figure 6—figure supplement 1*). Slowing does coincide with a shift in the forces defining

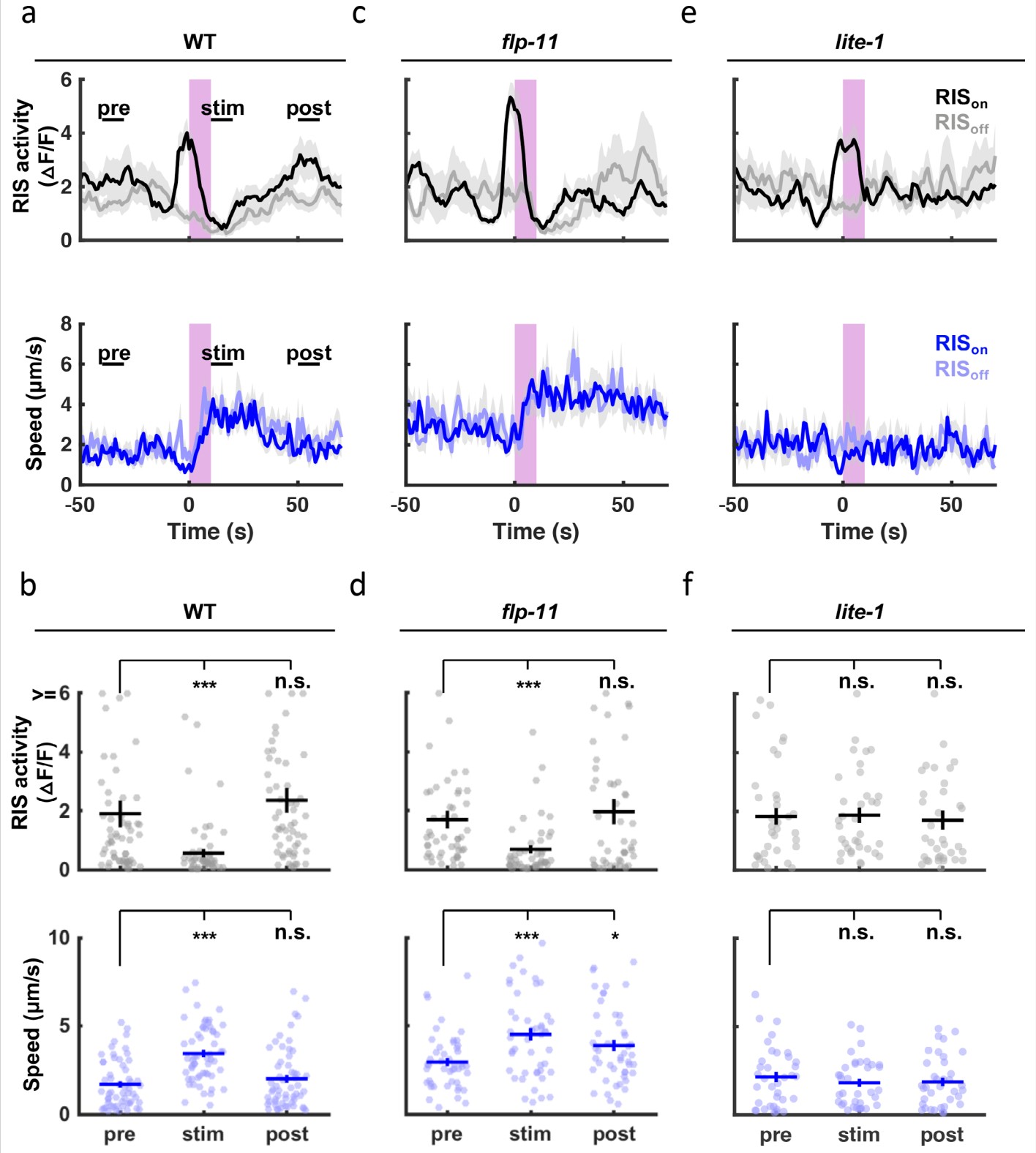

**Figure 9.** SWT quiescent bouts are reversible, but arousal responses are dampened by FLP-11. A brief (10 second) UVB light stimulus transiently inhibits RIS activity and evokes aroused embryo motion. UVB evoked RIS inhibition and aroused motion were both eliminated in *lite-1* mutants, which lack the endogenous light receptor. The arousal response is apparent even when the stimulus occurs during a quiescent bout (i.e. during an RIS transient), indicating that SWT quiescence is reversible. The arousal response is prolonged in *flp-11* mutants, suggesting that light responses are dampened by

*Figure 9 continued on next page*

*Figure 9 continued*

RIS during SWT. Normalized GCaMP intensity and speed of the RIS cell body were simultaneously recorded in individual embryos of the indicated genotypes before, during, and after a UVB stimulus (purple bar). Trials were categorized based on the state of RIS at the onset of the stimulus ($RIS_{on}$ and $RIS_{off}$). (**a, c, e**) Traces of RIS activity and speed are plotted (mean +/-SEM). Sample sizes are as follows: WT $RIS_{on}$, n=34; WT $RIS_{off}$, n=22; *flp-11* $RIS_{on}$, n=32; *flp-11* $RIS_{off}$, n=15; *lite-1* $RIS_{on}$, n=22; *lite-1* $RIS_{off}$, n=14 embryos. (**b, d, f**) Mean RIS activity and speed for the time points labeled in (**a**) ('pre', 'stim', 'post') are plotted. Each dot indicates an embryo. Mean +/-SEM. Data are shown for WT (**a, b**), *flp-11* (**c, d**), and *lite-1* mutants (**e, f**).'Pre', 'stim', and 'post' time points were compared using a non-parametric repeated measures test (Friedman), followed by a Dunn's multiple comparison test. Asterisks denote statistically distinguishable groups at $p < 0.001$ (***) or $p < 0.05$ (*). See *Figure 9—figure supplement 1* for data from brightfield recordings.

The online version of this article includes the following figure supplement(s) for figure 9:

**Figure supplement 1.** Arousal elicited by short-wavelength visible light.

body morphology, from a squeeze generated by contraction of circumferential actin bundles of the epidermal cytoskeleton to containment within a tough, yet flexible extracellular cuticle *Priess and Hirsh, 1986*. This structural transition could impact behavior; however, preliminary experiments (not shown) suggest that reduced motion is a consequence of decreased muscle activity rather than a structural constraint limiting motion. A shift from cytoskeletal to exoskeletal control of body shape is likely reiterated at each larval molt and could contribute to molt-associated lethargus quiescence *Raizen et al., 2008*. The brightfield motility assay promises to be a useful screening tool to further investigate this phenomenon.

Following the inactive phase, late-stage embryos exhibit a mature pattern of motion. Mature motion comprises sinusoidal crawling, prolonged bouts of forward and reverse motion, and a rhythmic pattern of brief quiescent bouts. All these features are grossly disrupted in *unc-13* mutants (*Figure 5a*), implying that they are driven by synaptic circuits. Our prior study suggests that bouts of directed motion are mediated by the forward and reversal locomotion circuits that operate post-hatching *Ardiel et al., 2017*.

## SWT, a rhythmic embryonic quiescence

We identify SWT as a rhythmic form of behavioral quiescence occurring late in embryogenesis. Osmotic stress and reduced insulin signaling also cause behavioral quiescence around hatching, which is accompanied by developmental arrest *Bayer et al., 2022*. Salt and insulin regulated quiescence and SWT quiescence occur at a similar stage of embryonic development and both require

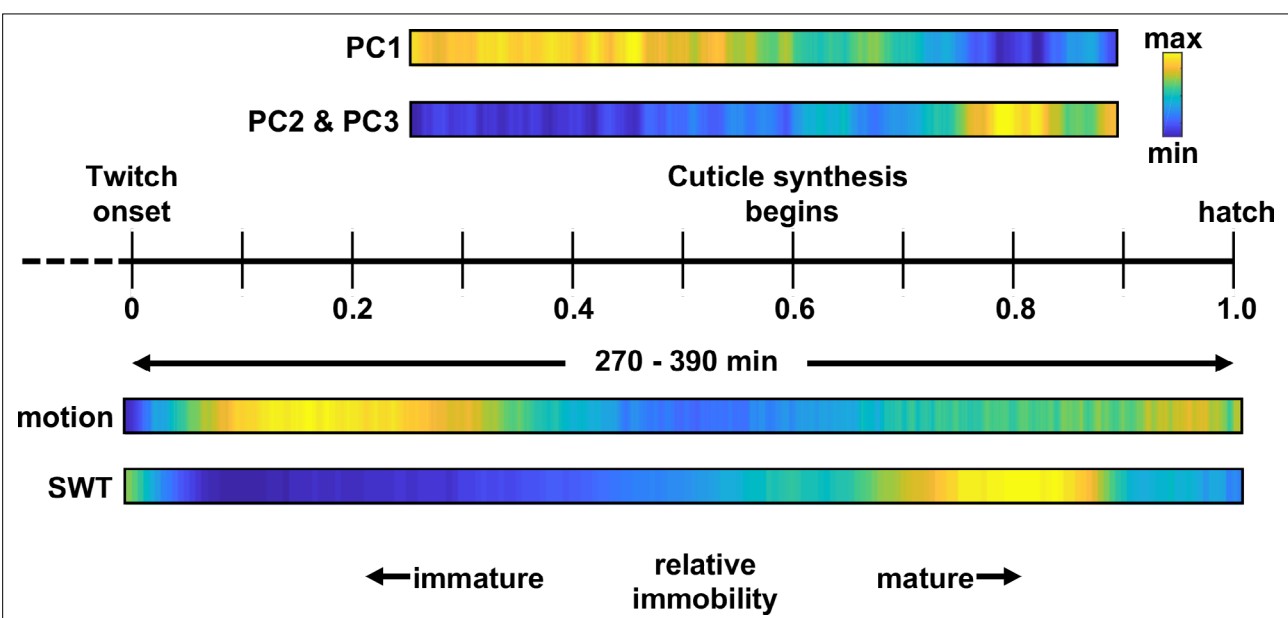

**Figure 10.** *C. elegans embryonic behavioral maturation*. Posture (top) and movement (bottom) data derived from seam cell tracking and brightfield recordings, respectively. Temporal scaling is shown along a relative timeline to account for day-to-day differences in developmental rate (*Figure 3—figure supplement 1*).

FLP-11, suggesting that these two forms of quiescence are mechanistically related. Other forms of *C. elegans* behavioral quiescence have been described during larval molts (lethargus) *Raizen et al., 2008* and in response to nutrient excess (satiety) *You et al., 2008*, cellular stress (e.g. heat shock) *Hill et al., 2014*; *Nelson et al., 2014*, starvation *Skora et al., 2018*; *Wu et al., 2018*, and physical restraint *Gonzales et al., 2019*. These quiescent states, particularly lethargus, have been shown to exhibit sleep-like properties: reversibility, reduced sensitivity to external stimuli, homeostatic regulation, and a stereotyped posture *Raizen et al., 2008*; *Choi et al., 2013*; *Iwanir et al., 2013*; *Cho and Sternberg, 2014*; *Tramm et al., 2014*; *Nagy et al., 2014*; *Schwarz and Bringmann, 2017*. Lethargus also shares deep molecular conservation with sleep in other systems, such as regulation by both PERIOD *Konopka and Benzer, 1971*; *Hardin et al., 1990*; *Sun et al., 1997*; *Tei et al., 1997*; *Monsalve et al., 2011* and AP-2 transcription factors *Mani et al., 2005*; *Turek et al., 2013*; *Kucherenko et al., 2016*; *Hu et al., 2020*.

Among the various quiescent states, SWT shares several properties with lethargus quiescence. First, both are promoted by RIS release of FLP-11 (*Figure 6*) *Turek et al., 2016*. Second, lethargus quiescence coincides with molting (i.e. cuticle replacement) and SWT peaks about an hour after the onset of cuticle synthesis (*Figure 10*). The timing of SWT relative to cuticle synthesis suggests that SWT may be similar to a late stage of lethargus. Consistent with this idea, quiescent bout durations in the second half of lethargus (≈10 s) *Iwanir et al., 2013* are similar to those seen in SWT (*Figure 8d*). Furthermore, the transcriptional profile around the time of SWT matches that of larval molting *Meeuse et al., 2020*. Finally, like lethargus and sleep, SWT quiescence can be reversed by an arousing light stimulus. Arousal is prolonged in *flp-11* mutants relative to controls (*Figure 9*, *Figure 9—figure supplement 1*), suggesting that responses to external stimuli are diminished during SWT. Despite these similarities, it is likely that further analysis will reveal important differences between SWT and lethargus. For example, during lethargus, stimulated arousal is followed by enhanced quiescence *Nagy et al., 2014*. This homeostatic response is a behavioral hallmark of sleep. We had conflicting results for homeostatic effects during SWT. Following arousal with visible light, we saw a small reduction in embryo motion below baseline (*Figure 9—figure supplement 1*), which could represent a homeostatic increase in quiescence. By contrast, we saw no evidence for homeostatic effects following the UVB stimulus (*Figure 9a and b*). This discrepancy could reflect differences in the arousing stimulus or differences in how embryo motion was tracked in the two assays (RIS tracking versus brightfield twitch profiles). A weak homeostatic drive is also consistent with SWT being analogous to a late stage of lethargus, when homeostatic regulation is greatly diminished *Nagy et al., 2014*.

Is SWT quiescence required for some aspect of early brain development? Correlated spontaneous activity is essential for the refinement and maturation of developing neuronal circuitry in a variety of contexts *Kirkby et al., 2013*. For example, following initial myogenic movements, behavioral maturation of *Drosophila* is dependent on patterned spontaneous neuronal activity in embryos *Crisp et al., 2008*; *Carreira-Rosario et al., 2021*; *Crisp et al., 2011*. Could RIS be influencing network connectivity by broadly structuring neuronal activity? Embryonic quiescence may also be a mechanism to preserve energy stores. In this scenario, disrupting sleep could predispose embryos to metabolic stress. Indeed, transcriptional stress responses are associated with sleep disruption in worms, flies, and mice *Sanders et al., 2017*; *Shaw et al., 2000*; *Cirelli, 2006*; *Terao et al., 2003*; *Naidoo et al., 2005*. It will be interesting to see if prenatal metabolic stress alters early brain development.

In summary, despite the presence of many important neurodevelopmental benchmarks, behavior during the final hours of *C. elegans* embryogenesis is largely unexplored. The embryonic posture library described here will provide a resource to test new hypotheses about how (and when) embryonic circuits assemble. Semi-automated tracking with MHHT should accelerate expansion of this resource, with the discovery of new embryonic behaviors providing an entry point to study nascent neuronal circuitry. The potential for recording pan-neuronal activity in unanesthetized embryos executing well-defined behaviors *Ardiel et al., 2017* as a stereotyped connectome emerges *Witvliet et al., 2021*, uniquely positions *C. elegans* to investigate fundamental aspects of early brain development.

## Materials and methods

Key resources table

| Reagent type (species) or resource | Designation | Source or reference | Identifiers | Additional information |
|---|---|---|---|---|
| Strain, strain background, (*C. elegans*) | *wIs51* | https://cgc.umn.edu/ | KP10922 | *wIs51* [SCMp::GFP +*unc-119*(+)] |
| Strain, strain background, (*C. elegans*) | *unc-13*(s69); *wIs51* | This paper | KP9135 | *Figure 5* |
| Strain, strain background, (*C. elegans*) | N2 Bristol | https://cgc.umn.edu/ | N2 | Wild-type reference |
| Strain, strain background, (*C. elegans*) | *unc-47*(e307) | https://cgc.umn.edu/ | CB307 | |
| Strain, strain background, (*C. elegans*) | *unc-17*(e245) | https://cgc.umn.edu/ | CB933 | |
| Strain, strain background, (*C. elegans*) | *acr-16*(ok789) | https://cgc.umn.edu/ | RB918 | |
| Strain, strain background, (*C. elegans*) | *unc-29*(x29) | Lewis et al., 1980 | KP7858 | |
| Strain, strain background, (*C. elegans*) | *eat-4*(ky5) | https://cgc.umn.edu/ | MT6308 | |
| Strain, strain background, (*C. elegans*) | *cat-1*(e1111) | https://cgc.umn.edu/ | CB1111 | |
| Strain, strain background, (*C. elegans*) | *egl-3*(n150) | https://cgc.umn.edu/ | MT150 | |
| Strain, strain background, (*C. elegans*) | *egl-21*(n476) | https://cgc.umn.edu/ | MT1071 | |
| Strain, strain background, (*C. elegans*) | *ceh-17*(np1) | https://cgc.umn.edu/ | IB16 | |
| Strain, strain background, (*C. elegans*) | *aptf-1*(gk794) | https://cgc.umn.edu/ | HBR227 | |
| Strain, strain background, (*C. elegans*) | *aptf-1*(tm3287) | https://cgc.umn.edu/ | HBR232 | |
| Strain, strain background, (*C. elegans*) | *flp-11*(tm2706) | https://cgc.umn.edu/ | HBR507 | |
| Strain, strain background, (*C. elegans*) | *unc-29*(x29); *nuTi233* | This paper | KP9744 | *nuTi233*[*pat-10* p::NLS-wCherry-NLS::SL2::UNC-29] |
| Strain, strain background, (*C. elegans*) | *nuTi580* | This paper | KP10749 | *nuTi580*[*flp-11* p::GCaMP6f:: mCherry] |
| Strain, strain background, (*C. elegans*) | *goeEx737* | https://cgc.umn.edu/ | HBR2256 | *goeEx737*[*flp-24* p::SL1:: GCaMP3.35::SL2::mKate2 +*unc-119*(+)] |
| Strain, strain background, (*C. elegans*) | *flp-11*(tm2706); *nuTi580* | This paper | KP10763 | *Figure 8* |
| Strain, strain background, (*C. elegans*) | *unc-25*(e156); *nuTi580* | This paper | KP10964 | *Figure 8—figure supplement 1* |
| Strain, strain background, (*C. elegans*) | *unc-25*(e156); *flp-11*(tm2706); *nuTi580* | This paper | KP10965 | *Figure 8—figure supplement 1* |
| Strain, strain background, (*C. elegans*) | *lite-1*(ce314); *nuTi580* | This paper | KP10762 | *Figure 9* |
| Strain, strain background, (*E. coli*) | OP50 | Brenner, 1974 | OP50 | Worm food |
| Recombinant DNA reagent | *pat-10* p::NLS-wCherry-NLS::SL2::UNC-29 | This paper | KP#4524 | UNC-29 expression in muscle |
| Recombinant DNA reagent | *flp-11* p::GCaMP6f:: mCherry | This paper | KP#4525 | GCaMP expression in RIS |
| Chemical compound, drug | poly-L-lysine | Sigma-Aldrich | P2636 | |

| Reagent type (species) or resource | Designation | Source or reference | Identifiers | Additional information |
|---|---|---|---|---|
| Software, algorithm | MATLAB | MathWorks | MATLAB R2018a | |
| Software, algorithm | MHHT | This paper, *Lauziere, 2022* | | https://github.com/lauziere/MHHT |
| Other | UVB LED | Marktech Optoelectronics | MTSM285 | *Figure 9* |

## Strains and reagents

Animals were maintained on nematode growth medium (NGM) seeded with *Escherichia coli* (OP50) as described in *Brenner, 1974*. The posture library was built using *wIs51*[*SCM*p::GFP +*unc-119*(+)], which expresses GFP in hypodermal seam cell nuclei. For the supplemental postures described in *Figure 4—figure supplement 1*, *wIs51* was combined with epidermal actin marker, *mcIS50*[*lin-26*p::vab-10(actin-binding domain)::GFP +*myo-2*p::GFP] and one of the following mCherry transgenes: *stIs10131*[*elt-7*p::H1-wCherry+*unc-119*(+)] (SLS20), *stIs10501*[*ceh-36*p::H1-wCherry::his-24*p::mCherry +*unc-119*(+)] (SLS21), *stIs10691*[*ref-1*p::H1-wCherry+*unc-119*(+)] (SLS22), *nuSi187*[*pat-10*p::NLS-GCaMP::mCherry-NLS] (SLS23), *stIs10138*[*tbx-38*p:: H1-wCherry; *unc-119*(+)] (SLS24), or *stIs10808*[*nob-1*p::H1-wCherry; *unc-119*(+)] (SLS25).

For rescuing *unc-29* expression in muscle, plasmid KP#4524 was generated by inserting the *pat-10* promoter (550 bp), the SV40 nuclear localization sequence, mCherry with introns, the EGL-13 nuclear localization sequence, the *gpd-2* 3' UTR with SL2 acceptor site, *unc-29* cDNA, and the *unc-54* 3' UTR between the SpeI and AgeI sites in pCFJ910.

For calcium imaging in RIS, plasmid KP#4525 was generated by inserting the *flp-11* promoter (3159 bp), GCaMP6f with introns, a flexible linker (GSSTSG(AP)$_7$ASEF), mCherry with introns, and the *unc-54* 3' UTR between the SbfI and StuI sites in pCFJ910.

The miniMOS method *Frøkjær-Jensen et al., 2014* was used to integrate KP#4524 and KP#4525, generating single copy insertions: *nuTi233*[*pat-10*p::NLS::mCherry::SL2::UNC-29] and *nuTi580*[*flp-11*p::GCaMP6f::mCherry].

Recordings were acquired at room temperature. Embryonic age was determined by measuring the time elapsed since twitch onset (defined as 430 mpf).

## Posture library

### Sample preparation

Gravid adults were dissected to release embryos, which were transferred to a patch of poly-L-lysine (1 mg/mL) in a diSPIM imaging chamber filled with M9 buffer. Embryo orientation was set to minimize Z-steps, that is, objective translation along the embryo's minor axis. Prior to fluorescence imaging, brightfield recordings were acquired to stage the embryos.

### Microscope

On a diSPIM *Kumar et al., 2014*, a pair of perpendicular water-dipping, long-working distance objectives (40 x, 0.8 NA) were used for brightfield and fluorescence imaging. Acquisition was controlled using Micro-Manager's diSPIM plugin (https://micro-manager.org/). Note that although the diSPIM can collect two orthogonal views, only a single view was used to build posture libraries, i.e., iSPIM mode. Sweeping a 488 *nm* laser beam (measured <1 μm after the objective), a two-dimensional MEMS mirror in the diSPIM scanhead generated the light-sheet and defined the image volume. To maximize speed, the camera (pco.edge 4.2 or ORCA-Flash 4.0) was oriented such that the chip readout direction was along the embryo's minor axis. The imaging objective was translated in step with the light-sheet, generating 36 planes at inter-plane spacing of 1.2 μm. Exposure time per plane was 4 ms and volumes were acquired at 3 Hz. To reduce file size, the camera was set to 4 x binning, resulting in 16-bit images with 0.65 μm pixels. See *Duncan et al., 2019* for detailed protocols on imaging embryos with the diSPIM.

For the supplemental postures described in *Figure 4—figure supplement 1*, diSPIM acquisition parameters were optimized for spatial resolution rather than speed. Specifically, diSPIM mode was used to acquire image volumes from perpendicular objectives with no pixel binning at the camera.

The two orthogonal views were then computationally fused to generate image volumes with near isotropic resolution *Wu et al., 2013*; *Guo et al., 2020*. Laser power was at least an order of magnitude greater than for the volumes acquired at 3 Hz (i.e. >30 $\mu W$, measured after the objective). The 561 nm laser line was scanned simultaneously, with dual-color image splitters separating detection at the camera (W-VIEW GEMINI, Hamamatsu). Volumes were acquired every 5 min from pre-twitching until hatching, with each image volume comprising 50 planes at a spacing of 1 $\mu m$. Exposure time per plane was 5 ms, with volumes acquired in <900 ms.

## Analysis

Image volumes were segmented using a large kernel 3D U-Net trained on an average of the Dice coefficient and binary cross-entropy loss functions. See below for additional details on image segmentation. In a volume with good detections, seam cells were manually positioned along the body axis by observing movement over several frames. From this seed volume, cell positions were tracked across time using MHHT. See below for additional details on MHHT.

Left and right seam cells were fit with natural cubic splines using MATLAB's 'cscvn' function. DV bends were then computed from the midpoint (mL/R) along the spline connecting adjacent seam cells and the corresponding midpoint (mR/L) on the opposite side of the body (*Figure 4a*). For example, for the DV bend between H0L and H1L:

- $\vec{v}_1$ connects mL to H0L;
- $\vec{v}_2$ connects mR to mL;
- $\vec{v}_3$ connects mL to H1L; and
- DV bend = $arctan2(([\vec{v}_1 \times \vec{v}_2] \times [\vec{v}_2 \times \vec{v}_3]) \cdot (\frac{\vec{v}_2}{|\vec{v}_2|}), [\vec{v}_1 \times \vec{v}_2] \cdot [\vec{v}_2 \times \vec{v}_3])$

The MATLAB class "msdanalyzer" was used to evaluate mean square displacement *Tarantino et al., 2014*.

## Brightfield motion assay

### Sample preparation

Several (≥4) gravid adults per strain were dissected directly in an M9 filled glass bottom microwell imaging dish (MatTek Corp., P35G-1.5–20 C). Up to 60 pre-twitching embryos were arrayed on a patch of poly-L-lysine (0.1 mg/mL) using water currents generated with a worm pick. Worm remains and unwanted embryos were removed with a micropipette and the dish was lidded.

### Microscope

Embryo arrays were recorded on an inverted microscope (Zeiss, Axiovert 100) with a 5x, 0.25 NA objective. A CCD camera (CoolSNAP HQ2) generated 16-bit images with a pixel size of 1.29 $\mu m$ Transmitted light power was adjusted to get background pixel counts of ≈500 from a 2 ms exposure. Images were acquired at 1 Hz using Micro-Manager control software.

### Analysis

Pixel intensities were extracted from a box (21x21 pixels) centered on each embryo and frame to frame pixel intensity changes larger than threshold (25) were tallied. The onset of sustained twitching was detected as the first tally (smoothed over 30 frames) greater than 120 of a potential 441 pixels. Background intensity for each frame was also extracted from a perimeter around the embryo (61x61 pixels). Hatching was detected as the point at which the mean intensity at the embryo position (smoothed over 30 frames) approximated the mean intensity of the background perimeter (signal-to-background ratio of at least 0.92). Fresh hatchlings occasionally swam into the field of view, registering pixel changes in the embryo and background. Data were therefore excluded for frames in which the background registered changes greater than threshold (35) in more than X pixels (40 of a potential 240 when smoothed over 30 frames).

Peak SWT power ratios were computed as follows:

1. For each embryo, 15 min twitch profiles were extracted with a sliding window (200 s step size) from 1 hr post twitch until hatching. Missing datapoints were replaced with a reversed duplication of the immediately preceding interval of the same length.

2. Using MATLAB's Fast Fourier transform ('fft') function, each 15 min twitch profile was transformed from the time domain (i.e. pixels changing per second, $y(t)$) to the frequency domain, $Y(f)$ for equally spaced frequency bins from $f_0$=0 to $f_{900}$=1 Hz, that is, the sampling rate.
3. For each 15-min twitch profile, the proportion of the signal attributed to the SWT frequency band was computed:

$$\text{SWT power ratio} = \frac{\sum_{20\,\text{mHz} \leq f \leq 40\,\text{mHz}} |Y(f)|^2}{\sum_{0 \leq f \leq 500\,\text{mHz}} |Y(f)|^2}$$

4. For each embryo, the timing and magnitude of the largest SWT power ratio was recorded.

To estimate a baseline for peak SWT power ratios (i.e. the dashed lines in *Figures 6d and 7a*), pairs of neighboring values were randomly shuffled prior to the discrete Fourier transformation (between steps 1 and 2 above) for a sample of wild-type embryos.

Scalograms were generated using MATLAB's Continuous 1-D wavelet transform ('cwt') function (with 'morse' wavelet).

Each genotype was evaluated in at least two independent experiments. Because coordinated aspects of motion are not discerned, all movements detected by the brightfield assay are referred to as twitches, regardless of the age of the embryo. The 'slow wave' terminology refers to the relatively low-frequency band (20–40 mHz) for which SWT is defined.

## Calcium Imaging
### Sample preparation
Several (>=15) gravid adults per strain were dissected directly in an M9 filled glass bottom microwell imaging dish (MatTek Corp., P35G-1.5–20 C). Embryos with no more than four cells were selected using currents generated with a worm pick and stuck to a patch of poly-L-lysine (0.1 mg/mL) in pairs, aligned end to end along their major axis (a configuration that enabled imaging two embryos per scan). Worm remains and unwanted embryos were removed with a micropipette and the dish was lidded. Prior to calcium imaging, the brightfield assay (see above) was used to time first twitch.

### Microscope
At least 215 min after the onset of twitching (i.e. ≈645 mpf), embryos were imaged on an inverted confocal microscope (Nikon, Eclipse Ti2) using a 20 x, 0.75 NA objective. Resonant scanning in two directions generated image planes of 1024x256 pixels (0.16 $\mu m$/pixel) for two channels (488 and 561 nm lasers) in 33.3 ms. Two embryos were imaged per scan. Volumes were acquired at 1 Hz for 5 min using a piezo stage stepping 7.875 $\mu m$ between 5 image planes (note that the pinhole was opened to minimize the number of steps required to span the embryos).

### Analysis
RIS cell body position and corresponding GCaMP and mCherry intensities were extracted from an image volume cropped around a single embryo. Specifically, following background subtraction (pixel counts-100), each image plane was filtered using a Gaussian smoothing kernel with standard deviation of 5 (MATLAB's 'imgaussfilt' function with 'symmetric' image padding to prevent edge artifacts). The position of the RIS cell body was identified in the mCherry channel as the brightest pixel of the filtered volume. The corresponding intensity in the filtered GCaMP channel was used to compute $\Delta R/R_0 = (R - R_0)/R_0$, where $R$ is the GCaMP/mCherry ratio and $R_o$ is the ratio at baseline, set as the 10th percentile for each trace. Calcium transients were identified in smoothed traces (moving mean with span 4) using an onset threshold of $\Delta R/R_0 \geq 2$ and an offset threshold of $\Delta R/R_0 < 1.5$. For ALA, only a single calcium transient was detected at the cell body in 5 min recordings from 42 embryos. However, calcium transients in ALA neurites were detectable using the mean pixel count of maximum intensity projections (onset threshold: $\Delta R/R_0 \geq 0.07$ and offset threshold: $\Delta R/R_0 < 0.05$).

## Seam cell nuclei detection
Traditional methods for blob detection such as Watershed *Falcão et al., 2004* and Laplacian of Gaussians (LoG) *Lindeberg, 1998*; *Lowe, 2004* were compared to a variety of deep-learning-based approaches *He et al., 2017*; *Çiçek et al., 2016*; *Weigert et al., 2020Weigert et al., 2020*; *Table 1*.

Each detection method yields a set of detected objects per image volume. A linear program was used to match detections to annotations at each frame of the annotation set. One-to-one matching aligned annotated nuclei centers and detection cluster centers per volume across the test set. The average precision, recall, and F1 score across test set volumes are reported in *Table 1*.

Given its superior performance, a 3D U-Net style convolutional neural network (CNN) was employed to perform semantic segmentation on the *C. elegans* embryo image volumes *Ronneberger et al., 2015*; *Çiçek et al., 2016*. The established 3D U-Net was augmented to use a size (5,5,5) kernel, as opposed to the original (3,3,3) kernel, extending the field of view at each layer. Strided 3D convolution layers downsample by a factor of two across lateral dimensions while preserving axial resolution. The limited axial resolution encodes more information per planar image due to the explicit downsampling occurring during imaging. The number of filters in each layer doubles from 16 to 32, 64, 128 when downsampling. The loss function is the sum of the binary cross-entropy and negative dice coefficient losses. The cross-entropy portion prioritizes accurate prediction of challenging individual voxels between close nuclei, but may produce noisy predictions. In juxtaposition, the dice coefficient prioritizes structural similarity in clusters of voxels constituting nuclei, but is known to consolidate sigmoid outputs at extreme values 0 and 1, limiting the model's ability to smoothly identify uncertainty in voxel prediction *Scherr et al., 2018*. Denote $\mathbf{y} \in \{0, 1\}^{X \times Y \times Z}$ and $\hat{\mathbf{y}} \in [0, 1]^{X \times Y \times Z}$ as the binary ground truth and sigmoid output tensors, respectively. The loss can then be written:

$$\mathcal{L}(\mathbf{y}, \hat{\mathbf{y}}) = \underbrace{-\sum_{i=1}^{X}\sum_{j=1}^{Y}\sum_{k=1}^{Z}[y_{ijk}\log(\hat{y}_{ijk}) + (1 - y_{ijk})(\log(1 - \hat{y}_{ijk}))]}_{\text{Cross Entropy}} + \underbrace{\frac{-2\sum_{i=1}^{X}\sum_{j=1}^{Y}\sum_{k=1}^{Z}y_{ijk}\hat{y}_{ijk}}{\sum_{i=1}^{X}\sum_{j=1}^{Y}\sum_{k=1}^{Z}(y_{ijk}^2 + \hat{y}_{ijk}^2)}}_{\text{Dice Coefficient}}$$

The model was trained via the Adam optimizer with an initial learning rate of $7.5 * 10^{-5}$ *Kingma and Ba, 2017*. The model yields image volumes of the same size as the input microscope image volumes, containing values $y_{ijk} \in [0, 1]$. Each volumetric output is thresholded conservatively (output voxels $y_{ijk} \geq 0.95$ are predicted to be part of a seam cell nucleus) in order to minimize incorrectly predicted voxels between apparently close nuclei, while simultaneously ensuring dim nuclei are captured by at least a few correctly predicted voxels. Supervoxels defined as disjoint clusters of the binarized output volume are returned as predicted instances of seam cell nuclei. The centroids of detected supervoxels serve as the detection set for each image volume.

## Multiple hypothesis hypergraph tracking

Each imaged embryo yields a sequence of ≈54000 image volumes which are processed in batch by our large kernel 3D U-Net. The CNN with post-processing produces detected objects in each image volume. The initial set of detections may contain nuclei clumped together and misclassified as a single detection, spurious voxels (false positives), or may miss dim nuclei (false negatives). Seam cell nuclei tracking is cast as a traditional multiple object tracking (MOT) problem; all seam cell nuclei positions together are referred to as the posture at each frame.

Multiple Hypothesis Tracking (MHT) *Reid, 2015*; *Blackman, 2018*; *Cox and Hingorani, 1996a* is the foremost paradigm for MOT, especially when tracking objects imaged with fluorescence microscopy *Feng et al., 2011*; *Chenouard et al., 2013*. MHT tracks objects independently by using linear motion models to describe object trajectories in a 'deferred decision' logic, leveraging smooth motion over several future frames to disentangle competing tracks. However, the sporadic twitches and rapid changes in frame-to-frame behavior preclude predictive modeling of seam cell nuclei locations. Also, MHT is unable to model relationships between objects in deciding track updates. The interdependence of seam cell nuclei movement inspired our proposed paradigm for flexibly integrating correlated object motion into MOT applications.

Multiple Hypothesis Hypergraph Tracking (MHHT) was developed as an extension of MHT to more effectively maintain an accurate representation of the *C. elegans* posture throughout late-stage embryogenesis by leveraging correlated seam cell motion. Relationships between seam cell nuclei were used to give enhanced context at the data association step, in which each detection is identified as a unique seam cell nucleus or designated debris. MHHT uses hypergraphs to characterize relationships between seam cells. Hypergraphs enable richer characterizations of object relationships than is possible with graphical approaches. Our best hypergraphical model considers simultaneous

assignments of many detections to nuclear identities to rank posture hypotheses. This model is then fit using seam cell nuclei coordinates from a second imaged embryo to weight hypergraphical features, further informing the quantification of frame-to-frame movements.

MHHT adapts hypothesis oriented MHT *Reid, 2015*; *Cox and Hingorani, 1996a* to include graphical interpolation for missed detections and hypergraphical evaluation of sampled hypotheses. In summary, Murty's algorithm identifies the *K* best solutions to the standard linear data association problem *Murty, 1968*; *Cox and Miller, 1996b*. This association problem, known as the gated Global Nearest Neighbor (GNN) approach, treats objects as independent, with motion confined to a limited region, referred to as a gate. The gate specifies which detections can be associated to each track; i.e, seam cell nuclei gates which contain no detections are reported missing, and will be graphically interpolated. Furthermore, a hypergraphical association function *f* measures the cost of the complete posture update: $\mathbf{Z}^{(t)}$. This process is iterated recursively across *N* future frames to contextualize the state update at time *t*. The exploration forms an exponentially growing search tree in which paths from the initial state $\mathbf{Z}^{(t-1)}$ expand into *K* solutions. Each of the *K* solutions following interpolation yields a hypothesized state $\mathbf{Z}(\hat{t}, k)$, initiating a recursion, adding to the cost of association at frame *t*. The states $\mathbf{Z}(\hat{t}, k)$ of the path with minimum summed hypergraphical association cost is chosen at frame *t*.

Anatomical constraints allow for physical models to contextualize embryonic movement. The models themselves are expressed as graphs or hypergraphs. *Figure 1c* depicts a graphical representation of an embryo used as a basis for modeling posture track updates. Edges appear along sides posterior to anterior, laterally between pairs of nuclei, and diagonally between sequential pairs. The first graphical association model, denoted *Embryo*, compares changes in edge lengths frame-to-frame to comprise the cost of the track update. Annotated data are used to estimate statistics of a parametric model to further describe embryonic behavior. Two such models, *Posture* and *Movement*, are explored to track posture. *Posture* is the data-driven enhancement of *Embryo*; the hypergraphical model measures the consistency in the shape of the embryo throughout successive frames. *Movement* is then a data enhanced version of the GNN filter, a graphical model evaluating patterned movement between nuclei.

The embryo graph $G = (V, E)$ specifies a set of edges *E* connecting seam cells locally. The attributed graph arising from established tracks $\mathbf{g}^{(t-1)} := \mathbf{g}(\mathbf{Z}^{(t-1)}; E)$ contextualizes the state $\mathbf{Z}^{(t-1)}$, serving as a basis for comparison across hypotheses at time *t*. Hypotheses evaluated at time *t*: $\hat{\mathbf{g}}^{(t)} := \mathbf{g}(\hat{\mathbf{Z}}^{(t)}; E)$ describe the posture of the embryo according to predicted states at time *t*. Frame-to-frame differences in the attributed representations are assumed multivariate Gaussian: $(\hat{\mathbf{g}}^{(t)} - \mathbf{g}^{(t-1)}) \sim \mathcal{N}(\mathbf{0}, \mathbf{\Sigma})$. The Mahalanobis distance *f* is used to evaluate a hypothesized state representation:

$$f(\hat{\mathbf{g}}^{(t)}, \mathbf{g}^{(t-1)}; \hat{\mathbf{\Sigma}}^{-1}) = \sqrt{(\hat{\mathbf{g}}^{(t)} - \mathbf{g}^{(t-1)})' \hat{\mathbf{\Sigma}}^{-1} (\hat{\mathbf{g}}^{(t)} - \mathbf{g}^{(t-1)})}$$

The covariance matrix $\mathbf{\Sigma}$ is estimated from a corpus of annotated data. States of all *n* objects from frames $t = 1, 2, \ldots, T$ are used as pairs $\{(\mathbf{Z}^{(1)}, \mathbf{Z}^{(2)}), (\mathbf{Z}^{(2)}, \mathbf{Z}^{(3)}), \ldots, (\mathbf{Z}^{(T-1)}, \mathbf{Z}^{(T)})\}$ to estimate frame-to-frame variation in pairs of hyperedge differences: $\{(\mathbf{g}^{(1)}, \mathbf{g}^{(2)}), (\mathbf{g}^{(2)}, \mathbf{g}^{(3)}), \ldots, (\mathbf{g}^{(T-1)}, \mathbf{g}^{(T)})\}$. Define $\bar{\mathbf{g}} = \frac{\sum_{t=1}^{T} \mathbf{g}^{(t+1)} - \mathbf{g}^{(t)}}{T-1}$. Then, the covariance matrix is estimated:

$$\hat{\mathbf{\Sigma}} := \frac{1}{T-1} \sum_{t=1}^{T} (\mathbf{g}^{(t+1)} - \mathbf{g}^{(t)} - \bar{\mathbf{g}})(\mathbf{g}^{(t+1)} - \mathbf{g}^{(t)} - \bar{\mathbf{g}})'$$

The *Embryo* and *Posture* models penalize associations which distort the embryo's shape. The bottom graph of *Figure 1c* highlights correlated changes in edge length relative to the V2L-V3L edge. Edges between track states vary in length as the embryo moves frame to frame. Denote the edges $E = [e_1, e_2, \ldots, e_M]$, where edge $e_j = (u_j, v_j)$ describes a relationship between nuclei $u_j$ and $v_j$. Each vector $\mathbf{e}_j^{(t)} = \mathbf{z}_{u_j}^{(t)} - \mathbf{z}_{v_j}^{(t)}$ describes the chord connecting states of nuclei $u_j$ and $v_j$ at time *t*. Then, the vector $\mathbf{E}^{(t)} = [\|\mathbf{e}_1^{(t)}\|_2, \|\mathbf{e}_2^{(t)}\|_2, \ldots, \|\mathbf{e}_M^{(t)}\|_2]$ describes lengths of these chords. Differences in chord lengths between frames $\mathbf{E}^{(t)} - \mathbf{E}^{(t-1)} \in R^{M \times 1}$ form the basis of the association cost. Then, the *Embryo* model is defined:

$$f_E(\hat{\mathbf{Z}}^{(t)}, \mathbf{Z}^{(t-1)}) = \sqrt{(\hat{\mathbf{E}}^{(t)} - \mathbf{E}^{(t-1)})' \mathbf{I} (\hat{\mathbf{E}}^{(t)} - \mathbf{E}^{(t-1)})} = \sum_{j=1}^{M} \sqrt{(\hat{\mathbf{E}}^{(t)} - \mathbf{E}^{(t-1)})^2} \tag{1}$$

Annotated data are used to estimate covariances between the $M$ differences across state updates. The resulting covariance matrix $\hat{\Sigma}_P$ scales differences in chord lengths among the $M$ chords present in $G$ to yield the *Posture* model:

$$f_P(\hat{\mathbf{Z}}^{(t)}, \mathbf{Z}^{(t-1)}; \hat{\Sigma}_P^{-1}) = \sqrt{(\hat{\mathbf{E}}^{(t)} - \mathbf{E}^{(t-1)})' \hat{\Sigma}_P^{-1} (\hat{\mathbf{E}}^{(t)} - \mathbf{E}^{(t-1)})} \tag{2}$$

Both *Embryo* and *Posture* are further characterized by the unary costs given by the gated GNN and the evaluated hypothesis $\varphi^{(t)}$.

*Movement* extends the traditional GNN cost to penalize *unnatural* movement between states. The distance between states at $t-1$ and $t$, $\sum_{i=1}^{n} \|\mathbf{z}_i^{(t)} - \mathbf{z}_i^{(t-1)}\|_2$, can be scaled by the inverse covariance matrix describing motion between pairs of nuclei. The states $\mathbf{z}_i^{(t)} = [x_i^{(t)}, y_i^{(t)}, z_i^{(t)}]$ and $\mathbf{z}_j^{(t-1)} = [x_j^{(t-1)}, y_j^{(t-1)}, z_j^{(t-1)}]$ can be expressed as element-wise differences:

$$\mathbf{Z}^{(t)} - \mathbf{Z}^{(t-1)} = \begin{bmatrix} \mathbf{z}_1^{(t)} \\ \mathbf{z}_1^{(t)} \\ \vdots \\ \mathbf{z}_n^{(t)} \end{bmatrix} - \begin{bmatrix} \mathbf{z}_1^{(t-1)} \\ \mathbf{z}_2^{(t-1)} \\ \vdots \\ \mathbf{z}_n^{(t-1)} \end{bmatrix} = \begin{bmatrix} x_1^{(t)} \\ y_1^{(t)} \\ z_1^{(t)} \\ x_2^{(t)} \\ y_2^{(t)} \\ z_2^{(t)} \\ \vdots \\ x_n^{(t)} \\ y_n^{(t)} \\ z_n^{(t)} \end{bmatrix} - \begin{bmatrix} x_1^{(t-1)} \\ y_1^{(t-1)} \\ z_1^{(t-1)} \\ x_2^{(t-1)} \\ y_2^{(t-1)} \\ z_2^{(t-1)} \\ \vdots \\ x_n^{(t-1)} \\ y_n^{(t-1)} \\ z_n^{(t-1)} \end{bmatrix} = \begin{bmatrix} x_1^{(t)} - x_1^{(t-1)} \\ y_1^{(t)} - y_1^{(t-1)} \\ z_1^{(t)} - z_1^{(t-1)} \\ x_2^{(t)} - x_2^{(t-1)} \\ y_2^{(t)} - y_2^{(t-1)} \\ z_2^{(t)} - z_2^{(t-1)} \\ \vdots \\ x_n^{(t)} - x_n^{(t-1)} \\ y_n^{(t)} - y_n^{(t-1)} \\ z_n^{(t)} - z_n^{(t-1)} \end{bmatrix} \in R^{3n \times 1} \tag{3}$$

Each pair of nuclei has an estimable $3 \times 3$ covariance matrix specifying the relationship between movement along each axis. The resulting block $3n \times 3n$ covariance matrix, $\Sigma_M$ then scales the difference between states:

$$f_M(\hat{\mathbf{Z}}^{(t)}, \mathbf{Z}^{(t-1)}; \hat{\Sigma}_M^{-1}) = \sqrt{(\hat{\mathbf{Z}}^{(t)} - \mathbf{Z}^{(t-1)})' \hat{\Sigma}_M^{-1} (\hat{\mathbf{Z}}^{(t)} - \mathbf{Z}^{(t-1)})} \tag{4}$$

The *Posture* and *Movement* models are combined additively to produce the *Posture-Movement* (*PM*) model. Readers are referred to https://github.com/lauziere/MHHT (copy archived at swh:1:rev:-f7e35a2e3ef398191b9e49a57f80d514d8f880c1) and *Lauziere et al., 2021*; *Lauziere, 2022* for further details on MHHT methodology.

## Acknowledgements

We thank the MBL Imaging Center, Whitman Fellows Program, and Neurobiology Course for supporting this project; Daniel Colón-Ramos, Zhirong Bao, Anthony Santella, Pavak Shah, Radu Balan, Hank Eden, Ghadi Salem, Richard Ikegami, Troy McDiarmid, Om Patange, and the Kaplan lab for valuable discussions. Strains were provided by the CGC (funded by NIH Office of Research Infrastructure Programs - P40 OD010440). The NIH and its staff do not endorse or recommend any company, product, or service.

## Additional information

### Funding

| Funder | Grant reference number | Author |
| --- | --- | --- |
| Hearst Foundations | | Evan L Ardiel |

| Funder | Grant reference number | Author |
|---|---|---|
| National Science Foundation | DGE-1632976 | Andrew Lauziere |
| National Institutes of Health | NS32196 | Joshua M Kaplan |
| National Institutes of Health | NS121182 | Joshua M Kaplan |
| National Institute of Biomedical Imaging and Bioengineering | | Hari Shroff |
| Howard Hughes Medical Institute | | Hari Shroff |

The funders had no role in study design, data collection and interpretation, or the decision to submit the work for publication.

## Author contributions

Evan L Ardiel, Andrew Lauziere, Conceptualization, Software, Formal analysis, Investigation, Methodology, Writing – original draft, Writing – review and editing; Stephen Xu, Software, Investigation, Methodology; Brandon J Harvey, Ryan Patrick Christensen, Investigation; Stephen Nurrish, Methodology; Joshua M Kaplan, Conceptualization, Supervision, Funding acquisition, Writing – review and editing; Hari Shroff, Conceptualization, Supervision, Methodology, Writing – review and editing

## Author ORCIDs

Evan L Ardiel http://orcid.org/0000-0002-9366-5751
Brandon J Harvey http://orcid.org/0000-0002-7471-9937
Stephen Nurrish http://orcid.org/0000-0002-2653-9384
Joshua M Kaplan http://orcid.org/0000-0001-7418-7179

## Decision letter and Author response

Decision letter https://doi.org/10.7554/eLife.76836.sa1
Author response https://doi.org/10.7554/eLife.76836.sa2

---

# Additional files

## Supplementary files

• Transparent reporting form

## Data availability

Annotated image volumes are available on FigShare. Code for MHHT is available on GitHub.

The following datasets were generated:

| Author(s) | Year | Dataset title | Dataset URL | Database and Identifier |
|---|---|---|---|---|
| Ardiel E | 2022 | Embryo image volumes | https://figshare.com/articles/journal_contribution/2017_04_06_zip/16725349 | FigShare, 16725349 |
| Ardiel E | 2022 | Embryo image volumes | https://figshare.com/articles/journal_contribution/2016_10_12_zip/16766788 | FigShare, 16766788 |
| Ardiel E | 2022 | Embryo image volumes | https://figshare.com/articles/journal_contribution/2017_04_03_zip/16821268 | FigShare, 16821268 |

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
