## [Editor Report]

Embryonic behavior is a widespread phenomenon that remains poorly understood in any model system. Recent developments in imaging, genetic, and computational tools allow for unprecedented analyses of motor behaviors, and the patterns of neuronal activity underlying embryonic development. Here, Ardiel et colleagues establish the roundworm *C. elegans* as a powerful system to study the developmental trajectories corresponding to embryonic behavior and to provide mechanistic insight into how late-stage bouts of activity are modulated.

---

## [Decision Letter]

**Decision letter after peer review:**

Thank you for submitting your article "Stereotyped behavioral maturation and rhythmic quiescence in *C. elegans* embryos" for consideration by *eLife*. Your article has been reviewed by 3 peer reviewers, and the evaluation has been overseen by a Reviewing Editor and Piali Sengupta as the Senior Editor. The reviewers have opted to remain anonymous.

Essential revisions:

The reviewers appreciate the technical novelty of the work. The data analysis and screening approaches have the potential to be adopted by many labs, and the rich behavioral dataset can serve as a foundation for numerous future studies. However, the enthusiasm for the work is dampened by several major concerns. While the identification of a previously unknown period of behavioral quiescence in the late embryo is exciting, its relationship with sleep remains largely speculative. Additional work will be necessary to support the central claims of the paper and to clarify the scope of the biological conclusions.

1. The authors should experimentally establish that the SWT is a rapidly reversible state, which is a key behavioral criteria for sleep. The link with sleep would still remain suggestive, as other behavioral criteria have not been established/tested (e.g., sleep homeostatic properties). This fact should be reflected in the Results and Discussion. Also, as discussed by reviewer #3, the results of the FLP-11 mutants calls into question whether SWT is the same as lethargus.

2. In the second half of the manuscript (high throughput analysis based on bright-field assay), the authors should examine whether the mutants that disrupt SWT have a defect in earlier embryonic behaviors. There is also an opportunity to address the function of the SWT bouts right after hatching. If successful, this extension could increase the broader significance of the study.

3. Given the broad readership of *eLife*, the authors should help non-experts better understand the technical aspects of the Multiple Hypothesis Hypergraph Tracking (MHHT) and what is learned biologically through the application of this technique. The authors should strive for broad accessibility of the MHHT section.

4. A new metric specific to movement should be introduced to discriminate between increases and decreases in movements that produce a change in the peak SWT power ratio.

5. The observations made with the light-sheet (Figures 2 and 3) and with bright-field (Figure 5) assays should be consolidated. Presently, both parts of the manuscript tend to read as disjointed.

6. Given the results presented in the manuscript, the discussion related to ASD appears to be far-fetched – it should be either removed or sharply reduced.

*Reviewer #1 (Recommendations for the authors):*

I have a few major concerns related to the metrics and nomenclature used to assigned phases and phenotypes.

1) The assignment of the observed phases could be more clear. Having a timeline-type schematic at the end of Figure 3 that combines the observations made in Figures2 and 3 will improve the current study and will be helpful for future studies in the field.

2) The authors should try to consolidate the observations made with the light-sheet (Figure 2 and 3) and with brightfield (Figure 5) as best as possible. Currently, it is not clear which exact phase SWT corresponds to from the phases described earlier. Does it completely coincide with sinusoidal waves or are parts of the observed sinusoidal waves episodic SWT-like and some not?

3) While a change in the peak SWT power ratio is useful to reveal a loss of periodicity, it fails to show the underlying cause. A decrease (unc-13 mutant, Figure 5b) or increase (flp-11 mutant, Figure 6b) in movement will give the same peak SWT power ratio change as compared to WT. A metric that directly speaks to movement will improve the paper (e.g. cumulative prop. pixel binned by x time). This will be particularly useful for the mutants shown on Figure 5e that are not followed up on in the next figure such as egl-3 and egl-21.

*Reviewer #2 (Recommendations for the authors):*

Oddly, the authors describe SWT briefly in the context of their initial analysis without highlighting its novelty or identifying it as the SWT phenomenon. This comes later, when the brightfield assay allows them to show it is an unc-13-dependent phenomenon. In general, the most interesting new discovery could be described in more detail ("directional persistence" is a little vague when Figure 3e differentiates between forward and backward activity).

Some of the language describing the behavior is a little opaque. What is the definition of "twitching," "flipping," and "coiling"? In particular, I'm uncertain about the choice of "slow wave twitch." Why "twitch"? It looks like mostly a prolonged bout of forward movement, not a "twitch." This is particularly problematic since the term "twitch" is often used to describe the uncoordinated muscle contractions of mid-embryogenesis. I really think slow wave twitch is the wrong name. To my mind, a twitch is a kind of spasm-not the coordinating forward movement that appears to be the case here. In general, the nature of the SWT behavior could be described a bit more clearly.

I'd like to see a bit more detail on the SWT behavior. Figure 3e gives the clearest representation of what the behavior looks like, but if you have a time-lapse or another representation that would really help the reader visualize the behavior. At the very least, "SWT" or whatever it is to be called, should be introduced in the paragraph of lines 183-197 when it is first described, rather than two sections later. As written, it's a bit too descriptive, without highlighting the novelty of the behavior at the time when it is most clearly described.

I think I understood how the peak power ratio provides a mathematical description of SWT, but I wonder if the authors could provide a couple sentences somewhere in lines 221-237 explaining how this serves as a proxy for identifying the behavior, which would be helpful to non-experts.

Figure 5c shows SWT at around 700 mpf, but Figure 6b shows it at around 740 mpf. The authors carefully describe this as a phenomenon that is within the last 1-2 hours, but I wonder why the signature is appearing at different times. Do we think this represents some natural plasticity? Differences in experimental circumstance? Imprecision in defining mpf from one experiment to the next?

At what temperature were the analyses performed? How was temperature kept uniform? This matters in mpf reckoning with other published descriptions of embryonic development.

The study makes some speculations on the potential function of SWT that are not terribly persuasive without some additional experimental work. One does wonder about the health and welfare of mutant L1's that fail to go through SWT. It would be really interesting to know if there were behavioral or physiological consequences for flp-11 or aptf-1 mutant L1's. Of course, they won't sleep well at the first larval molt, complicating matters pretty quickly. But if one could document increased stress response, lowered viability after starvation, or some other L1 characteristic, this might help point to a functional role for SWT.

The authors make the connection to the energy-intensive process of cuticle synthesis, as is the case for larval sleep bouts, but never show where cuticle synthesis actually occurs. The timing of cuticle synthesis should be declared in the text and indicated in Figure 1a. I believe that doing so will show that cuticle synthesis takes place a bit before the SWT phenomenon. This does not undercut the argument, but does distinguish it a bit from larval bouts where quiescence precedes molting and cuticle synthesis.

Those looking for clinical significance might like the ASD connection, but I don't find the argument hugely persuasive at the moment. RIS is "among the most enriched" in ASD gene expression? Perhaps you can be a bit more precise-is it among the top 5 neurons?

*Reviewer #3 (Recommendations for the authors):*

Might most significant concerns with the manuscript, which lead me to question appropriateness for *eLife*, are:

1. The first half of the paper is very highly technical, and I think just too specific for such a broad journal. I am not sure there are ways to simplify/rewrite to fix this issue

2. There is really only 1 biological result here – the late embryonic quiescent/SWT behavior. Aside from a few mutant/molecular experiments, the authors do not take this very far. If the goal of the paper is a focus on this behavior, I think more depth would be required, specifically with regard to behavioral experiments like reversibility.

3. The authors do not go far enough to show that SWT is a new form of behavioral quiescence. The FLP-11 result is shaky, given the strong persistent negative correlation between RIS activity and motion.

4. The discussion of sleep and autism should be removed. It leads readers to believe more is shown in this data than really is present.

---

## [Author Response]

Essential revisions:The reviewers appreciate the technical novelty of the work. The data analysis and screening approaches have the potential to be adopted by many labs, and the rich behavioral dataset can serve as a foundation for numerous future studies. However, the enthusiasm for the work is dampened by several major concerns. While the identification of a previously unknown period of behavioral quiescence in the late embryo is exciting, its relationship with sleep remains largely speculative. Additional work will be necessary to support the central claims of the paper and to clarify the scope of the biological conclusions.

In the revised manuscript, we provide new data further supporting a link between SWT quiescence, lethargus, and sleep. These results are briefly summarized here and are described in greater detail below in responses to specific reviewer comments. We now show that: (1) SWT quiescence can be reversed by aversive light stimuli (Figure 9a-b; Figure 9-Supp 1); (2) sensory responses are dampened during SWT (Figure 9c-d; Figure 9-Supp 1); and (3) that for at least one stimulus, arousal is followed by a brief period of inhibited motion, potentially consistent with a homeostatic drive (Figure 9-Supp. 1). In the discussion, we speculate that SWT is analogous to a late stage of lethargus:

lines 413-421:

“Among the various quiescent states, SWT shares several properties with lethargus quiescence. First, both are promoted by RIS release of FLP-11 (Figure 7, Figure 8; Turek et al., 2016). Second, lethargus quiescence coincides with molting (i.e. cuticle replacement) and SWT peaks about an hour after the onset of cuticle synthesis (Figure 10). The timing of SWT relative to cuticle synthesis suggests that SWT may be similar to a late stage of lethargus. Consistent with this idea, quiescent bout durations in the second half of lethargus (~10 s, Iwanir et al., 2013) are similar to those seen in SWT (Figure 8d). Furthermore, the transcriptional profile around the time of SWT matches that of larval molting (Meeuse et al., 2020). Finally, like lethargus and sleep, SWT quiescence can be reversed by an arousing light stimulus.”

Despite these new data/revisions, we remain agnostic as to whether SWT qualifies as a bona fide sleep state. The central claim we hope to convey is that SWT represents a neuronally controlled embryonic behavior amenable to high-throughput mutant analysis.

1. The authors should experimentally establish that the SWT is a rapidly reversible state, which is a key behavioral criteria for sleep. The link with sleep would still remain suggestive, as other behavioral criteria have not been established/tested (e.g., sleep homeostatic properties). This fact should be reflected in the Results and Discussion.

New experiments and text were added to address these concerns:

Reversibility of SWT quiescent bouts, lines 328-344:

“Other forms of quiescence are associated with diminished responsiveness to arousing stimuli. To see if SWT shares this property, we analyzed the behavioral response to light. In adults and larvae, short-wavelength light has been shown to elicit arousal via endogenous photoreceptor LITE-1 (Edwards et al., 2008; Liu et al., 2010; Gong et al., 2017). Here, we monitored motion and activity of the RIS cell body following a UVB light pulse (285 nm). In these experiments, a 10 second irradiance was delivered during peak SWT (~1h before hatching). Stimulus trials were retrospectively identified as those occurring when RIS was active (RIS_on_) or inactive (RIS_off_). UVB elicited behavioral arousal in embryos (Figure 9a, b). Rapid and robust arousal was apparent in both RIS_off_ and RIS_on_ trials (Figure 9a), confirming that SWT quiescence was reversible. The duration of UVB evoked behavioral arousal was increased in flp-11 mutants compared to control embryos (Figure 9a-d). Together, these results suggest that FLP-11 dampens responses to UVB during the peak SWT period. Coincident with behavioral arousal, UVB irradiance evoked an acute inhibition of RIS activity (Figure 9a-d), as previously shown with blue light in larvae (Wu et al., 2018). Given the negative correlation between RIS activation and head speed (Figure 8f), inhibition of RIS likely contributes to the behavioral arousal elicited by UVB. Both UVB evoked behavioral arousal and RIS inhibition were eliminated in mutants lacking the LITE-1 photoreceptor (Figure 9e, f).”

Dampened sensory responses and potential homeostatic regulation of SWT, lines 345-353:

“To further investigate the impact of SWT on sensory responses, we used the brightfield twitch assay to monitor behavioral responses to longer wavelength visible light. Similar to UVB, visible light evoked an arousal response that was prolonged in flp-11 mutants (Figure 9-Supp. 1). Thus, arousal responses to aversive light stimuli are exaggerated in mutants lacking SWT quiescence. Unlike UVB, arousal elicited by visible light was followed by motion slightly (albeit significantly) lower than baseline (Figure 9-Supp. 1). This inhibited motion was lost in flp-11 mutants and may represent a homeostatic response to the arousing stimulus. Collectively, these results suggest that SWT quiescence is reversible and that behavioral responses to arousing stimuli are dampened by FLP-11 release during SWT.”

In the Discussion, we now evaluate evidence for homeostatic regulation of SWT, lines 423-432:

“….it is likely that further analysis will reveal important differences between SWT and lethargus. For example, during lethargus, stimulated arousal is followed by enhanced quiescence. This homeostatic response is a behavioral hallmark of sleep. We had conflicting results for homeostatic effects during SWT. Following arousal with visible light, we saw a small reduction in embryo motion below baseline (Figure 9-Supp. 1), which could represent a homeostatic increase in quiescence. By contrast, we saw no evidence for homeostatic effects following the UVB stimulus (Figure 9a, b). This discrepancy could reflect differences in the arousing stimulus or differences in how embryo motion was tracked in the two assays (RIS tracking versus brightfield twitch profiles). A weak homeostatic drive is also consistent with SWT being analogous to a late stage of lethargus, when homeostatic regulation is greatly diminished (Nagy, 2014).”

Also, as discussed by reviewer #3, the results of the FLP-11 mutants calls into question whether SWT is the same as lethargus.

In the revised text, we show that *flp-11* mutations significantly reduce quiescence elicited by RIS activation (Figure 8g). The magnitude of the *flp-11* lethargus quiescence defect strongly depends on the measure used to assess motion. In lethargus, FLP-11 primarily inhibits motion of the anterior end of the animal. Motion of the larval body (assessed by centroid motion) during lethargus remains largely quiescent in *flp-11* mutants [Robinson et al., Micropub Bio 2019]. Thus, FLP-11 only partially accounts for quiescence in both SWT and lethargus. For these reasons, we believe that FLP-11 could play similar roles in SWT and lethargus.

Prompted by this comment, we now provide a more detailed comparison of SWT and lethargus quiescence:

lines 413-432:

“Among the various quiescent states, SWT shares several properties with lethargus quiescence. First, both are promoted by RIS release of FLP-11 (Figure 6; Turek et al., 2016). Second, lethargus quiescence coincides with molting (i.e., cuticle replacement) and SWT peaks about an hour after the onset of cuticle synthesis (Figure 10). The timing of SWT relative to cuticle synthesis suggests that SWT may be similar to a late stage of lethargus. Consistent with this idea, quiescent bout durations in the second half of lethargus (~10 s, Iwanir et al., 2013) are similar to those seen in SWT (Figure 8d). Furthermore, the transcriptional profile around the time of SWT matches that of larval molting (Meeuse et al., 2020). Finally, like lethargus and sleep, SWT quiescence can be reversed by an arousing light stimulus. Arousal is prolonged in flp-11 mutants relative to controls (Figure 9, Figure 9-Supp. 1), suggesting that responses to external stimuli are diminished during SWT. Despite these similarities, it is likely that further analysis will reveal important differences between SWT and lethargus. For example, during lethargus, stimulated arousal is followed by enhanced quiescence. This homeostatic response is a behavioral hallmark of sleep. We had conflicting results for homeostatic effects during SWT. Following arousal with visible light, we saw a small reduction in embryo motion below baseline (Figure 9-Supp. 1), which could represent a homeostatic increase in quiescence. By contrast, we saw no evidence for homeostatic effects following the UVB stimulus (Figure 9a, b). This discrepancy could reflect differences in the arousing stimulus or differences in how embryo motion was tracked in the two assays (RIS tracking versus brightfield twitch traces). A weak homeostatic drive is also consistent with SWT being analogous to a late stage of lethargus, when homeostatic regulation is greatly diminished (Nagy, 2014).”

2. In the second half of the manuscript (high throughput analysis based on bright-field assay), the authors should examine whether the mutants that disrupt SWT have a defect in earlier embryonic behaviors.

We are confused by this comment. An *unc-13* mutation (which blocks neuropeptide release, neurotransmitter release, and SWT) has no apparent effect on immature embryo behavior (Figures 5b-d); consequently, it seems unlikely that other SWT deficient mutants (e.g. *flp-11*) could alter early behavior. Please let us know if we have misinterpreted this comment. Nonetheless, we address this comment by presenting entire twitch profiles for SWT mutants (Figure 6-Supp. 1).

There is also an opportunity to address the function of the SWT bouts right after hatching. If successful, this extension could increase the broader significance of the study.

We agree that assessing the function of SWT is interesting and this is the focus of our ongoing work. We hope that the reviewers will agree that evaluating functional or developmental defects in SWT mutants goes beyond the scope of the current paper and can be addressed in a future study.

3. Given the broad readership of eLife, the authors should help non-experts better understand the technical aspects of the Multiple Hypothesis Hypergraph Tracking (MHHT) and what is learned biologically through the application of this technique. The authors should strive for broad accessibility of the MHHT section.

We thank the reviewers for this comment. To address this concern, we have essentially rewritten our opening Results section, “*Building embryo posture libraries*”. Our hope is that these extensive revisions will help non-experts better understand the challenges addressed by MHHT.

Revisions are as follows:

lines 65-111:

“Building posture libraries to investigate embryo behavior proved challenging for several reasons. […] MHHT should improve tracking compared to strategies where nuclei are assumed to move independently.”

4. A new metric specific to movement should be introduced to discriminate between increases and decreases in movements that produce a change in the peak SWT power ratio.

In response to this comment (and the related comment #12), we would first emphasize that increases or decreases in movement alone cannot account for changes in the peak SWT power ratio. For example, scaling WT movement amplitudes up or down has no effect on SWT power ratio. Instead, SWT power ratios detect changes in the temporal structure of motion, as shown by the dramatically reduced SWT power ratios produced when analyzing randomly shuffled movement data (see SWT power ratio baselines in Figure 6d and Figure 7a). This logic is now clearly stated in the revised text:

lines 273-276:

“Although motility was generally decreased in *unc-17* mutants (Figure 6-Supp. 1), changes in the magnitude of motion alone cannot explain changes in the relative power of the SWT frequency band. The SWT power ratio quantifies the temporal structure of motion.”

Having said that, we agree that when SWT defects are found, further analysis is required to understand the nature of those defects. We now show full twitch profiles for the SWT mutants presented in Figure 6d (formerly Figure 5e; Figure 6-Supp. 1a). As suggested by reviewer #1 (comment #12), we also plot cumulative distribution functions binned by age (Figure 6-Supp. 1b; Figure 7-Supp. 1).

Following up on the SWT defect in *flp-11* mutants, we use a metric that directly assesses motion (RIS cell body displacement).

lines 313-325:

“Behavioral quiescence was substantially reduced but not eliminated in *flp-11* mutants (Figure 8c, d). […] These results suggest that a prominent feature of late-stage embryonic behavior is a rhythmic pattern of quiescence elicited by two RIS neurotransmitters, FLP-11 and GABA.”

5. The observations made with the light-sheet (Figures 2 and 3) and with bright-field (Figure 5) assays should be consolidated. Presently, both parts of the manuscript tend to read as disjointed.

Prompted by this comment, we rearranged the manuscript and now use both motion assays throughout the manuscript. In the revised text, the brightfield assay and SWT are introduced early on (Figure 3), before even describing postural analysis based on seam cell tracking (Figure 4). We also include a new summary figure which consolidates the observations made with the light-sheet and bright-field assays (Figure 10).

6. Given the results presented in the manuscript, the discussion related to ASD appears to be far-fetched – it should be either removed or sharply reduced.

As suggested, all discussion of ASD has been removed from the manuscript.

Reviewer #1 (Recommendations for the authors):1) The assignment of the observed phases could be more clear. Having a timeline-type schematic at the end of Figure 3 that combines the observations made in Figures2 and 3 will improve the current study and will be helpful for future studies in the field.

As suggested, we include a new figure (Figure 10) summarizing the timeline of the various behavioral phases, and other embryonic developmental landmarks.

2) The authors should try to consolidate the observations made with the light-sheet (Figure 2 and 3) and with brightfield (Figure 5) as best as possible. Currently, it is not clear which exact phase SWT corresponds to from the phases described earlier. Does it completely coincide with sinusoidal waves or are parts of the observed sinusoidal waves episodic SWT-like and some not?

As detailed above (see response to comment #8), we addressed this concern by extensively rearranging the manuscript so that observations made with the light-sheet and brightfield assays are described in parallel. The brightfield assay and SWT are introduced earlier on (Figure 3), before even describing postural analysis based on seam cell tracking (Figure 4). As mentioned above, we also include a new summary figure for this purpose (Figure 10).

3) While a change in the peak SWT power ratio is useful to reveal a loss of periodicity, it fails to show the underlying cause. A decrease (unc-13 mutant, Figure 5b) or increase (flp-11 mutant, Figure 6b) in movement will give the same peak SWT power ratio change as compared to WT. A metric that directly speaks to movement will improve the paper (e.g. cumulative prop. pixel binned by x time). This will be particularly useful for the mutants shown on Figure 5e that are not followed up on in the next figure such as egl-3 and egl-21.

In response to this comment (which is similar to comment #7), we would first emphasize that increases or decreases in movement alone cannot account for changes in the peak SWT power ratio. For example, scaling WT movement amplitudes up or down has no effect on SWT power ratio. Instead, SWT power ratios detect changes in the temporal structure of motion, as shown by the dramatically reduced SWT power ratios produced when analyzing randomly shuffled movement data (see SWT power ratio baselines in Figure 6d and Figure 7a). This logic is now clearly stated in the revised text:

lines 273-276:

“Although motility was generally decreased in *unc-17* mutants (Figure 6-Supp. 1), changes in the magnitude of motion alone cannot explain changes in the relative power of the SWT frequency band. The SWT power ratio quantifies the temporal structure of motion.”

Having said that, we agree that when SWT defects are found, further analysis is required to understand the nature of those defects. We now show full twitch profiles for the SWT mutants presented in Figure 6d (formerly Figure 5e; Figure 6-Supp. 1a). As suggested by reviewer #1 (comment #12), we also plot cumulative distribution functions binned by age (Figure 6-Supp. 1b; Figure 7-Supp. 1).

Following up on the SWT defect in *flp-11* mutants, we use a metric that directly assesses motion (RIS cell body displacement).

lines 313-325:

“Behavioral quiescence was substantially reduced but not eliminated in *flp-11* mutants (Figure 8c, d). Using the onset of calcium transients to align behavior and fluorophore intensities (GCaMP and mCherry), we found that the GCaMP signal (and not the mCherry signal) was negatively correlated with head speed (Figure 8f), confirming that RIS activation was associated with behavioral slowing. This relationship persisted in *flp-11* mutants, however compared to wild-type, the behavioral slowdown was more transient (Figure 8f) and less likely to lead to a pause (Figure 8g). The residual behavioral slowing found in *flp-11* mutants could be mediated by another RIS neurotransmitter, e.g., GABA. Consistent with this idea, compared to *flp-11* single mutants, the behavioral slowdown associated with RIS calcium transients was diminished in *flp-11;unc-25* GAD double mutants, which are deficient for GABA synthesis (Figure 8-Supp. 1). These results suggest that a prominent feature of late-stage embryonic behavior is a rhythmic pattern of quiescence elicited by two RIS neurotransmitters, FLP-11 and GABA.”

Reviewer #2 (Recommendations for the authors):Oddly, the authors describe SWT briefly in the context of their initial analysis without highlighting its novelty or identifying it as the SWT phenomenon. This comes later, when the brightfield assay allows them to show it is an unc-13-dependent phenomenon.

As detailed above (response to comment #8), the text was revised to present brightfield and seam cell tracking data together, earlier in the manuscript:

lines 156-170:

“Seam cell tracking has two potential limitations as a strategy to analyze embryo behavior. First, the fluorescence imaging required for posture tracking could artifactually distort the observed embryonic behaviors (e.g., due to subtle phototoxic effects). Second, although our imaging pipeline is semi-automated, seam cell tracking remains labor intensive. To address these concerns, we devised an independent high throughput brightfield assay to assess overall embryonic motility. Brightfield images of up to 60 embryos were simultaneously acquired at 1 Hz, and frame-to-frame changes in pixel intensity were used as a proxy for embryo movement (Figure 3a, Figure 3-Video 1). The resulting twitch profiles were highly stereotyped, exhibiting three salient phases: an immature active phase (~450-550 mpf), followed by a relatively inactive period (~550-650 mpf), followed by a second active phase (>~650 mpf; Figure 3b) with a broadened distribution of movement magnitudes (Figure 3c). We generated scalograms to visualize the temporal structure of twitch profiles over a range of timescales. This analysis revealed a prominent feature in the 20-40 mHz frequency band, occurring about an hour before hatching (Figure 3d). This feature was attributed to an increased propensity for prolonged pausing, as had been seen in seam cell tracks at this stage (Figure 2g). We call this behavioral signature slow wave twitch (SWT).”

In general, the most interesting new discovery could be described in more detail

Prompted by this comment, we added several new experiments characterizing the SWT quiescent bouts:

a) We include a new video (Video 1) highlighting pausing in posture space.

lines 302-304:

“To further describe SWT behavior, we analyzed embryo postures at the developmental stage corresponding to peak SWT behavior. We found that pauses occurred throughout posture space (Video 1), suggesting that SWT quiescent bouts are not associated with specific body postures.”

b) Rather than a cross-correlation, we now show the behavioral slowing associated with RIS activation.

lines 314-319:

“Using the onset of calcium transients to align behavior and fluorophore intensities (GCaMP and mCherry), we found that the GCaMP signal (and not the mCherry signal) was negatively correlated with head speed (Figure 8f), confirming that RIS activation was associated with behavioral slowing. This relationship persisted in *flp-11* mutants, however compared to wild-type, the behavioral slowdown was more transient (Figure 8f) and less likely to lead to a pause (Figure 8g).”

c) Analyzing *unc-25* GAD mutants, we show that the residual slow-down associated with RIS activation depends on GABA.

lines 319-326:

“The residual behavioral slowing found in *flp-11* mutants could be mediated by another RIS neurotransmitter, e.g., GABA. Consistent with this idea, compared to *flp-11* single mutants, the behavioral slowdown and quiescence associated with RIS calcium transients was diminished in *flp-11;unc-25* GAD double mutants, which are deficient for GABA synthesis (Figure 8-Supp. 1). These results suggest that a prominent feature of late-stage embryonic behavior is a rhythmic pattern of quiescence elicited by two RIS neurotransmitters, FLP-11 and GABA. GABA release was associated with transient behavioral slowing, whereas FLP-11 release was essential for sustained pausing.”

d) We recorded calcium transients in a second sleep-promoting neuron (ALA) during SWT.

lines 307-310:

“Tracking GCaMP/mCherry ratios in the RIS cell body in embryos aged at least 645 mpf (Figure 8a, b), we observed RIS calcium transients at a rate within the SWT frequency band (24.8 +/- 1.6 mHz, mean +/- SEM; Figure 8c). In contrast, ALA neuron calcium transients occurred far less frequently (1.9 +/- 0.3 mHz, mean +/- SEM).”

e) We monitored sensory responses to light stimuli during SWT.

lines 328-341:

“Other forms of quiescence are associated with diminished responsiveness to arousing stimuli. To see if SWT shares this property, we analyzed the behavioral response to light. In adults and larvae, short-wavelength light has been shown to elicit arousal via endogenous photoreceptor LITE-1 (Edwards et al., 2008; Liu et al., 2010; Gong et al., 2017). Here, we monitored motion and activity of the RIS cell body following a UVB light pulse (285 nm). In these experiments, a 10 second irradiance was delivered during peak SWT (~1h before hatching). Stimulus trials were retrospectively identified as those occurring when RIS was active (RISon) or inactive (RISoff). UVB elicited behavioral arousal in embryos (Figure 9a, b). Rapid and robust arousal was apparent in both RISoff and RISon trials (Figure 9a), confirming that SWT quiescence was reversible. The duration of UVB evoked behavioral arousal was increased in flp-11 mutants compared to control embryos (Figure 9a-d). Together, these results suggest that FLP-11 signalling dampens responding to UVB even during SWT motile bouts. Coincident with behavioral arousal, UVB irradiance evoked an acute inhibition of RIS activity (Figure 9a-d), as previously shown with blue light in larvae (Wu et al., 2018).”

("directional persistence" is a little vague when Figure 3e differentiates between forward and backward activity).

Prompted by this comment, the phrase “directional persistence” was removed from the manuscript.

lines 142-154:

“To further characterize embryo behavior, we considered speed and movements along the anteroposterior axis…. Timing movements along the anteroposterior axis, we found that the duration of both forward and backward bouts increased over the final two hours of embryogenesis, as did the duration of pausing (Figure 2e,g). In summary, seam cell tracking reveals a consistent developmental progression in embryo behavior. Immature embryos move nearly continuously in short trajectories, while mature embryos show prolonged bouts of forward and backward movement punctuated with pausing.”

Some of the language describing the behavior is a little opaque. What is the definition of "twitching," "flipping," and "coiling"?

These terms are now defined in the text as follows:

lines 558-560:

“Because coordinated aspects of motion are not discerned, all movements detected by the brightfield assay are referred to as twitches, regardless of the age of the embryo”

lines 240-242:

“… flips were defined as transitions between dorsal and ventral coils, i.e. from fully dorsally to fully ventrally bent postures and vice versa.”

lines 190-191:

“PC1 captures ventral or dorsal coiling (i.e. all ventral or all dorsal body bends, respectively)”

In particular, I'm uncertain about the choice of "slow wave twitch." Why "twitch"? It looks like mostly a prolonged bout of forward movement, not a "twitch." This is particularly problematic since the term "twitch" is often used to describe the uncoordinated muscle contractions of mid-embryogenesis. I really think slow wave twitch is the wrong name. To my mind, a twitch is a kind of spasm-not the coordinating forward movement that appears to be the case here.

The SWT signal is defined using the bright field assay, where pixel intensity changes are used as a proxy for motion. Because coordinated aspects of motion are not discerned in the brightfield assay, just overall motion, we refer to the output of these assays as twitch profiles. The “slow wave” terminology refers to the fact that the behavior is defined by a relatively low frequency band (20-40 mHz). This is all now explained explicitly in the revised text:

lines 558-561:

“Because coordinated aspects of motion are not discerned, all movements detected by the brightfield assay are referred to as twitches, regardless of the age of the embryo. The “slow wave” terminology refers to the relatively low frequency band (20-40 mHz) for which SWT is defined.”

For these reasons, we prefer to retain the SWT terminology. Authors may prefer different terms to describe their data. In our defense, we point out that SWT and twitch signals are explicitly (i.e., mathematically) defined in the text; consequently, readers will not be confused about the meaning of these terms.

In general, the nature of the SWT behavior could be described a bit more clearly.I'd like to see a bit more detail on the SWT behavior. Figure 3e gives the clearest representation of what the behavior looks like, but if you have a time-lapse or another representation that would really help the reader visualize the behavior.

We now provide extensive new data analyzing SWT behavior (see response to comment #18). As requested, we now provide a video (Video 1) highlighting pausing in posture space:

lines 302-304:

“To further describe SWT behavior, we analyzed embryo postures at the developmental stage corresponding to peak SWT behavior. We found that pauses occurred throughout posture space (Video 1), suggesting that SWT quiescent bouts are not associated with specific body postures.”

At the very least, "SWT" or whatever it is to be called, should be introduced in the paragraph of lines 183-197 when it is first described, rather than two sections later. As written, it's a bit too descriptive, without highlighting the novelty of the behavior at the time when it is most clearly described.

As detailed in response to comment #8, we rearranged the manuscript considerably to consolidate the observations made with the light-sheet and brightfield assays. We now introduce the brightfield assay and SWT earlier on (Figure 3), before even describing postural analysis based on seam cell tracking (Figure 4). We also include a new summary figure for this purpose (Figure 10).

We thank the reviewer for this suggestion. However, we prefer the to retain the narrative strategy employed in our revised manuscript. We believe that this structure best describes the overall progression of embryonic behavior, the detailed aspects of the rhythmic SWT quiescence that emerges in late embryos, and how the posture library can be used to analyze newly discovered behaviors.

Our justification for retaining this narrative structure is as follows. As noted above (response to comment #21), SWT is defined with the brightfield twitch assay. There are hints of these quiescent bouts in the seam cell tracking data, but precisely when they occur, their duration, their disruption by mutations, and their correlation with RIS activity is all detailed in the latter parts of the manuscript. These details cannot be included earlier in the manuscript. Finally, this narrative structure provides an example to illustrate how readers can use the posture library in the future. Using independent assays to identify and localize a new embryonic behavior (SWT in our case), readers can refer to the posture library provided here to infer what specific postures and behavioral motifs could correspond to that signal.

I think I understood how the peak power ratio provides a mathematical description of SWT, but I wonder if the authors could provide a couple sentences somewhere in lines 221-237 explaining how this serves as a proxy for identifying the behavior, which would be helpful to non-experts.

We clarified this in the Results section:

lines 261-263:

“To quantify SWT, we scanned the twitch profile of each embryo for the 15 minutes in which 20-40 mHz most dominated the power spectrum. The SWT power ratio refers to the proportion of the power spectrum accounted for by the 20-40 mHz frequency band.”

Figure 5c shows SWT at around 700 mpf, but Figure 6b shows it at around 740 mpf. The authors carefully describe this as a phenomenon that is within the last 1-2 hours, but I wonder why the signature is appearing at different times. Do we think this represents some natural plasticity? Differences in experimental circumstance? Imprecision in defining mpf from one experiment to the next?

We added new analysis evaluating day-to-day variability in the time from twitch onset to hatch (Figure 3-Supp. 1). As shown in this figure, the timing of SWT relative to the onset of twitching and hatching is consistent across days. We also now comment on variability in the text of the Results section:

lines 171-181:

“While the general pattern of behavioral maturation was highly stereotyped, absolute timings of the different phases varied between experiments. For example, the mean time from first twitch to hatch ranged from 4.5 to 6.5 hours (324.1 +/- 37.0 minutes, mean +/- SEM) across 31 brightfield recordings (Figure 3-Supp. 1). This variability could result from differences in temperature or buffer salinity, both of which have been shown to influence the rate of development (Wood, 1988; Atakan et al., 2020). The poly-L-lysine used for sticking embryos to the coverslip is an additional potential source of variability. However, the timing of SWT scaled with hatch time (Figure 3-Supp. 1), suggesting that the relative timing of development was consistent across experiments. In summary, similar motion profiles were observed in the brightfield and seam cell tracking assays, further suggesting that the observed progression of behaviors is an authentic feature of embryonic development.”

At what temperature were the analyses performed? How was temperature kept uniform? This matters in mpf reckoning with other published descriptions of embryonic development.

Experiments were run at room temperature, which is now indicated in the Methods (line 475). A new summary figure (Figure 10) uses temporal scaling along a relative timeline to facilitate conversions to mpf.

The study makes some speculations on the potential function of SWT that are not terribly persuasive without some additional experimental work. One does wonder about the health and welfare of mutant L1's that fail to go through SWT. It would be really interesting to know if there were behavioral or physiological consequences for flp-11 or aptf-1 mutant L1's. Of course, they won't sleep well at the first larval molt, complicating matters pretty quickly. But if one could document increased stress response, lowered viability after starvation, or some other L1 characteristic, this might help point to a functional role for SWT.

The potential importance of SWT post-hatching is the focus of ongoing work. We appreciate the interest and propose that this would be better (and more fully) addressed in a future study.

The authors make the connection to the energy-intensive process of cuticle synthesis, as is the case for larval sleep bouts, but never show where cuticle synthesis actually occurs. The timing of cuticle synthesis should be declared in the text and indicated in Figure 1a. I believe that doing so will show that cuticle synthesis takes place a bit before the SWT phenomenon. This does not undercut the argument, but does distinguish it a bit from larval bouts where quiescence precedes molting and cuticle synthesis.

We now include a summary figure (Figure 10) illustrating the timing of the onset of cuticle synthesis and SWT behavior. Indeed, the onset of cuticle synthesis precedes peak SWT. Based on this timing, we propose in the discussion that SWT may be akin to a late stage of lethargus:

lines 413-420:

“Among the various quiescent states, SWT shares several properties with lethargus quiescence. First, both are promoted by RIS release of FLP-11 (Figure 7, Figure 8; Turek et al., 2016). Second, lethargus quiescence coincides with molting (i.e. cuticle replacement) and SWT peaks about an hour after the onset of cuticle synthesis (Figure 10). The timing of SWT relative to cuticle synthesis suggests that SWT may be similar to a late stage of lethargus. Consistent with this idea, quiescent bout durations in the second half of lethargus (~10 s, Iwanir et al., 2013) are similar to those seen in SWT (Figure 8d). Furthermore, the transcriptional profile around the time of SWT matches that of larval molting (Meeuse et al., 2020).”

Those looking for clinical significance might like the ASD connection, but I don't find the argument hugely persuasive at the moment. RIS is "among the most enriched" in ASD gene expression? Perhaps you can be a bit more precise-is it among the top 5 neurons?

As suggested, we removed all discussion of ASD from the manuscript.